# Joint distribution of currents in the symmetric exclusion process

**Aurélien Grabsch[1][⋆], Pierre Rizkallah[2] and Olivier Bénichou[1]**

**1** Sorbonne Université, CNRS, Laboratoire de Physique Théorique de la
Matière Condensée (LPTMC), 4 Place Jussieu, 75005 Paris, France
**2** Sorbonne Université, CNRS, Physicochimie des Electrolytes et Nanosystèmes
Interfaciaux (PHENIX), 4 Place Jussieu, 75005 Paris, France

⋆ aurelien.grabsch@sorbonne-universite.fr

## Abstract

The symmetric simple exclusion process (SEP) is a paradigmatic model of diffusion in a single-file geometry, in which the particles cannot cross. In this model, the study of currents have attracted a lot of attention. In particular, the distribution of the integrated current through the origin, and more recently, of the integrated current through a moving reference point, have been obtained in the long time limit. This latter observable is particularly interesting, as it allows to obtain the distribution of the position of a tracer particle. However, up to now, these different observables have been considered independently. Here, we characterise the joint statistical properties of these currents, and their correlations with the density of particles. We show that the correlations satisfy closed integral equations, which generalise the ones obtained recently for a single observable. We also obtain boundary conditions verified by these correlations, which take a simple physical form for any single-file system. As a consequence of our results, we quantify the correlations between the displacement of a tracer, and the integrated current of particles through the origin.



# 1 Introduction

The Symmetric Exclusion Process (SEP) is a paradigmatic model of single-file diffusion [1,2], which has been the object of several recent and important developments [3–8]. In this model, particles perform symmetric random walks in continuous time on an infinite one-dimensional lattice, with the constraint that there can be at most one particle per site. In this context, several quantities have attracted attention: (i) the integrated current through the origin $Q_t$ (defined as the number of particles which have crossed the origin from left to right, minus those from right to left, up to time $t$) [8–11]; (ii) the position $X_t$ of a tracer [5–7, 12–18], initially placed at the origin; (iii) the generalised current $J_t$ which counts the number of particles which cross a moving boundary[1] at position $x_t$ (counted positively from left to right, and negatively from right to left) [3, 4, 12]. This latter observable actually provides an alternative way to study the displacement of a tracer, since its position $X_t$ corresponds to the value of $x_t$ for which $J_t = 0$, because the order of the particles is conserved [3, 4, 12].

The statistical properties of the current $Q_t$ have been fully characterised by the computation of its cumulant generating function using Bethe ansatz [9]. Concerning the position $X_t$ of a tracer, its fluctuations have first been quantified by the computation of the variance [12]. The full distribution has later been computed, first in the high-density limit [19], then in the low-density limit [16, 17, 20, 21], and finally at arbitrary density [3, 4] by relying on tools from integrable probabilities (which give a microscopic solution). These latter works [3, 4] actually provide the full cumulant generating function of the generalised current $J_t$, from which the statistical properties of the position of the tracer is deduced.

Recently, combining microscopic and macroscopic approaches, it has been shown that, in the long time limit, all these results can be easily recovered from the solution of a simple integral equation [6, 7]. This equation is satisfied by generalized density profiles [5, 6], which

---

[1]This observable is also called a *height function* since it is involved in a classical mapping between exclusion processes and interface models [3, 4].

characterise the correlations between the observable under consideration ($Q_t$, $J_t$ or $X_t$) and the density of particles in the SEP. On top of providing a more direct way to obtain the statistical properties of these observables, this equation constitutes a strikingly simple closure of the infinite hierarchy of equation satisfied by these generalized density profiles [6,7]. These results are part of a context of intense activity around exact solutions for one-dimensional interacting particle systems [8,22–25].

Although the individual properties of $Q_t$, $J_t$ or $X_t$ have been characterised, the determination of their correlations remains an open question. These observables are indeed expected to be strongly correlated since, for instance, if the tracer (initially at the origin) moves to a position $X_t$ to the right, then the current $Q_t$ through the origin can only be positive. The determination of these correlations is the main goal of this article.

Here, relying only on a macroscopic description of the SEP, we show that the integral equation of [6,7] can be generalised to describe the joint correlations between the currents $Q_t$, $J_t$ and the density of particles of the SEP in the long time limit. As a consequence, we deduce the joint statistical properties of $Q_t$ and $J_t$, and, as a byproduct, those of $Q_t$ and $X_t$. Importantly, this equation is completed by boundary conditions, which were derived from microscopic considerations in [5–7]. Here, we provide a macroscopic derivation of these boundary relations, and extend them beyond the SEP to any single-file system. Furthermore, thanks to this generalisation, we give a clear physical meaning to these relations.

The article is organised as follows. We first present in Section 2 a summary of our main results, followed by a discussion of these results and their consequences in Section 2.3. We then present in Section 3 the Macroscopic Fluctuation Theory (MFT) [26–28], which gives a large scale description of diffusive systems, and is our starting point. In Section 4 we first illustrate on the simpler case of low density how this approach can be used to study joint properties of different observables. We give in Section 5 the derivation of our main results, which rely both on the MFT and on the inverse scattering technique [29] which has recently been applied to MFT and related problems [8,22–25]. We give in Section 6 a perturbative solution of our main equations, from which the first joint cumulants of $Q_t$, $J_t$ and $X_t$ are obtained. We finish by several concluding remarks in Section 7.

## 2 Summary of the main results

We consider a SEP in which particles hop with rate 1. We describe the state of the system by the set of occupation numbers $\{\eta_i(t)\}_{i \in \mathbb{Z}}$, with $\eta_i(t) = 1$ if site $i$ is occupied at time $t$, and 0 otherwise. Initially, we consider that each site is filled independently with probability $\rho_+$ for $i > 0$ and $\rho_-$ for $i \leq 0$.

The integrated current $Q_t$ counts the total number of particles that cross the origin from left to right, minus the number from right to left, up to time $t$. It can be written explicitly in terms of the occupation numbers by comparing the number of particles to the right of the origin at times $t$ and 0,

$$Q_t = \sum_{r>0} (\eta_r(t) - \eta_r(0)) \,. \tag{1}$$

Similarly, the generalised current $J_t$ counts the number of particles that cross a fictitious moving boundary located at $x_t$ at time $t$. It can be expressed in terms of the occupation numbers, by comparing the number of particles to the right of $x_t$ at time $t$, and the number of particles to the right of $x_0 = 0$ at initial time. Explicitly, it can be written as

$$J_t = \sum_{r>x_t} [\eta_r(t) - \eta_r(0)] - \sum_{r=1}^{x_t} \eta_r(0), \tag{2}$$

which can equivalently be expressed in a slightly different manner by subtracting the mean density at infinity $\rho_+$ and separating the sums.

This gives the definition

$$J_t = \sum_{r > x_t} (\eta_r(t) - \rho_+) - \sum_{r > 0} (\eta_r(0) - \rho_+) - \rho_+ x_t, \quad x_t = \lfloor \xi \sqrt{t} \rfloor. \tag{3}$$

We have chosen a specific expression for $x_t$, such that $x_t \sim \sqrt{t}$ for large $t$ since the system is diffusive. We will consider only the case $\xi > 0$, but the case $\xi < 0$ can be obtained similarly.

Our main results concern the joint cumulant generating function of the two currents $Q_t$ (1) and $J_t$ (3), in the long time limit

$$\psi_\xi(\lambda, \nu, t) = \ln \left\langle e^{\lambda Q_t + \nu J_t} \right\rangle \underset{t \to \infty}{\simeq} \sqrt{t} \, \hat{\psi}_\xi(\lambda, \nu), \tag{4}$$

and their correlation with the density of surrounding particles, encoded in the generalised density profiles [5]

$$w_r(\lambda, \nu, t) = \frac{\left\langle \eta_r(t) \, e^{\lambda Q_t + \nu J_t} \right\rangle}{\left\langle e^{\lambda Q_t + \nu J_t} \right\rangle} \underset{t \to \infty}{\simeq} \Phi\left( x = \frac{r}{\sqrt{t}} \right). \tag{5}$$

The profile $\Phi$ also depends on $\lambda$, $\nu$ and $\xi$, but we omit the dependency on these variables to simplify the notations. These profiles actually correspond to the average occupations of the sites in the so-called *tilted* or *canonical* path ensemble [30–32]. This ensemble describes the dynamics of the system under a bias due to the exponential terms in (5), which favors the realisations in which $Q_t$ and $J_t$ differ from their expected values. The profiles $w_r$, and thus $\Phi$, measure the response of the density of particles to this bias. In particular, if for instance $\lambda > 0$, the exponential in (5) favors realisations in which $Q_t$ is larger than its average. In these realisations, we expect an accumulation of particles to the right of the origin, and a depletion of the left, due to the particles that have crossed the origin from left to right. This is indeed what we observe in Fig. 1, left. The same interpretation holds for $J_t$. Furthermore, expanding (5) in powers of $\lambda$ and $\nu$, $w_r$ generates all the correlations $\left\langle \eta_r(t) Q_t^n J_t^m \right\rangle_c$ between the currents $Q_t$, $J_t$ and the occupations $\eta_r(t)$. In particular we discuss below the lowest order correlations.

We also characterise the correlations between the currents and the initial density of particles, encoded in the initial profile

$$\bar{w}_r(\lambda, \nu, t) = \frac{\left\langle \eta_r(0) \, e^{\lambda Q_t + \nu J_t} \right\rangle}{\left\langle e^{\lambda Q_t + \nu J_t} \right\rangle} \underset{t \to \infty}{\simeq} \bar{\Phi}\left( x = \frac{r}{\sqrt{t}} \right), \tag{6}$$

which has a similar interpretation to the profile at final time (5). For instance, for $\lambda > 0$, the configurations favored by the exponential are the ones which yield a larger current $Q_t$. At $t = 0$, this is realised by a fluctuation of the initial condition with more particles on the left of the origin, and less on the right. Then, the dynamics will make these particles flow through the origin, yielding a large current $Q_t$. This is what is shown in Fig. 1, right.

We have obtained equations satisfied by the profiles $\Phi$ and $\bar{\Phi}$, from which the cumulant generating function $\hat{\psi}_\xi$ is deduced. We first present these main equations, before giving some of their consequences on the correlations between the different observables and finally discussing the status of these results and relations with the recent works [6, 7].

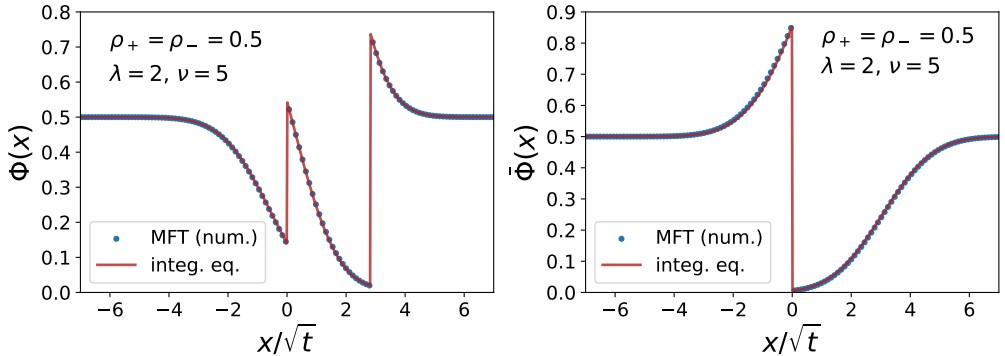

Figure 1: Profiles $\Phi$ (left) and $\bar{\Phi}$ (right) obtained from the numerical resolution of the integral equations (8,9) with the conditions (11-17) (solid red lines), compared to the numerical resolution of the MFT equations (blue points). Here, we have set $\xi = 2\sqrt{2}$, $\lambda = 2$ and $\nu = 5$. These can be interpreted as a mean density profile, under the condition that $Q_t = 0.83$ and $J_t = 0.27$. The numerical schemes used to obtain these curves are described in Appendix C and Appendix D.

## 2.1 Equations for the correlations and the cumulants

Instead of the profiles at final time $\Phi$ and initial time $\bar{\Phi}$, we found that it is their derivatives which verify closed equations. More precisely, we define the functions

$$\Omega(x) \equiv \begin{cases} a_- \Phi'(x), & \text{for } x < 0, \\ a_0 \Phi'(x), & \text{for } 0 < x < \xi, \\ a_+ \Phi'(x), & \text{for } x > \xi, \end{cases} \qquad \bar{\Omega}(x) \equiv \begin{cases} b_- \bar{\Phi}'(x), & \text{for } x < 0, \\ b_+ \bar{\Phi}'(x), & \text{for } x > 0, \end{cases} \quad (7)$$

with multiplicative constants $a_-$, $a_0$, $a_+$, $b_-$ and $b_+$ to be determined. We find that the functions $\Omega$ and $\bar{\Omega}$ satisfy the following integral equations

$$\Omega(x) + \alpha \int_0^\infty \Omega(y)\Omega(x-y)\Theta(y-x)\mathrm{d}y + \beta \int_\xi^\infty \Omega(y)\Omega(x+\xi-y)\Theta(y-x)\mathrm{d}y$$

$$+ \alpha\beta \int_\xi^\infty \mathrm{d}y \int_0^\xi \mathrm{d}z\, \Omega(y)\Omega(z)\Omega(x+\xi-y-z)\Theta(y+z-x-\xi) = K(x), \quad (8)$$

$$\bar{\Omega}(x) + \int_0^\infty \bar{\Omega}(y)\bar{\Omega}(x-y)\Theta(y-x)\mathrm{d}y = \bar{K}(x), \quad (9)$$

where $\Theta$ is the Heaviside step function, and the kernels

$$K(x) = \frac{e^{-x^2/4}}{\sqrt{4\pi}}, \quad \bar{K}(x) = \alpha\frac{e^{-x^2/4}}{\sqrt{4\pi}} + \beta\frac{e^{-(x-\xi)^2/4}}{\sqrt{4\pi}} + \alpha\beta \int_0^\xi \Omega(y)\frac{e^{-(x+y-\xi)^2/4}}{\sqrt{4\pi}}\mathrm{d}y, \quad (10)$$

which involve additional parameters $\alpha$ and $\beta$. The multiplicative constants in the definitions (7) are determined by the following boundary conditions

$$\mu(\Phi(0^+)) - \mu(\Phi(0^-)) = \lambda, \quad \mu(\Phi(\xi^+)) - \mu(\Phi(\xi^-)) = \nu, \quad (11)$$

$$[\partial_x \mu(\Phi)]_{0^-}^{0^+} = 0, \quad [\partial_x \mu(\Phi)]_{\xi^-}^{\xi^+} = 0, \quad (12)$$

$$\mu(\bar{\Phi}(0^+)) - \mu(\bar{\Phi}(0^-)) = -(\lambda + \nu) + \mu(\rho_+) - \mu(\rho_-), \quad [\partial_x \mu(\bar{\Phi})]_{0^-}^{0^+} = 0, \quad (13)$$

which involve the chemical potential $\mu$. For the SEP, it takes the simple form

$$\mu(\rho) = -\ln\left(\frac{1}{\rho} - 1\right). \tag{14}$$

We also have boundary conditions at infinity,

$$\lim_{x\to\pm\infty} \Phi(x) = \rho_{\pm}, \quad \lim_{x\to\pm\infty} \bar{\Phi}(x) = \rho_{\pm}. \tag{15}$$

The last constants $\alpha$ and $\beta$ are determined by conservation equations

$$\int_{-\infty}^{\infty} [\Phi(x) - \bar{\Phi}(x)]\mathrm{d}x = 0. \tag{16}$$

$$\int_{-\infty}^{\infty} \bar{K}(x)\mathrm{d}x = \alpha + \beta + \alpha\beta \int_{0}^{\xi} \Omega(y)\mathrm{d}y = \omega, \tag{17}$$

where

$$\omega = \rho_{+}(e^{-\lambda-\nu} - 1) + \rho_{-}(e^{\lambda+\nu} - 1) + \rho_{+}\rho_{-}(e^{\lambda+\nu} - 1)(e^{-\lambda-\nu} - 1), \tag{18}$$

coincides with the single parameter identified in the SEP [9,10,33], but with two parameters $\lambda$ and $\nu$.

Together, Eqs.(8-16) fully determine the profiles $\Phi$ and $\bar{\Phi}$. Their knowledge allows to deduce the joint cumulant generating function by using (see below)

$$\partial_{\lambda}\hat{\psi}_{\xi} = \int_{0}^{\infty} [\Phi(x) - \bar{\Phi}(x)]\mathrm{d}x, \quad \partial_{\nu}\hat{\psi}_{\xi} = \int_{0}^{\infty} [\Phi(x+\xi) - \bar{\Phi}(x)]\mathrm{d}x - \rho_{+}\xi. \tag{19}$$

## 2.2 Consequences on the observables

We only present the results for the case of a flat initial density $\rho_{+} = \rho_{-} = \rho$ because they take a simpler form, but the results from the general case $\rho_{+} \neq \rho_{-}$ can be obtained similarly. From the above equations, we recover the known results on $Q_t$ [9] and $J_t$ [4], for instance

$$\lim_{t\to\infty} \frac{1}{\sqrt{t}} \langle Q_t \rangle = 0, \quad \lim_{t\to\infty} \frac{1}{\sqrt{t}} \langle J_t \rangle = -\rho\xi, \tag{20}$$

$$\lim_{t\to\infty} \frac{1}{\sqrt{t}} \langle Q_t^2 \rangle = \frac{2\rho(1-\rho)}{\sqrt{\pi}}, \quad \lim_{t\to\infty} \frac{1}{\sqrt{t}} \langle J_t^2 \rangle_c = \rho(1-\rho)\left(\frac{2e^{-\frac{\xi^2}{4}}}{\sqrt{\pi}} + \xi\,\mathrm{erf}\left(\frac{\xi}{2}\right)\right), \tag{21}$$

where $\mathrm{erf}(z) = \frac{2}{\sqrt{\pi}}\int_{0}^{z} e^{-x^2}\mathrm{d}x$. In addition, we obtain the joint statistical properties of these two observables, such as their covariance

$$\lim_{t\to\infty} \frac{1}{\sqrt{t}} \langle Q_t J_t \rangle_c = \rho(1-\rho)\left(\frac{1+e^{-\frac{\xi^2}{4}}}{\sqrt{\pi}} - \frac{\xi}{2}\,\mathrm{erfc}\left(\frac{\xi}{2}\right)\right), \tag{22}$$

where $\mathrm{erfc}(z) = 1 - \mathrm{erf}(z)$ is the complementary error function. This shows that $Q_t$ and $J_t$ are strongly correlated for $\xi \to 0$, and their correlation decay when $\xi \to \infty$, as

$$\lim_{t\to\infty} \frac{\langle Q_t J_t \rangle}{\sqrt{\langle Q_t^2 \rangle \langle J_t^2 \rangle}} \simeq \begin{cases} 1 - \frac{\xi}{2}\sqrt{\frac{\pi}{2}}, & \text{for } \xi \to 0, \\ \dfrac{1}{2^{3/4}\pi^{1/4}\sqrt{\xi}}, & \text{for } \xi \to \infty. \end{cases} \tag{23}$$

Remarkably, these behaviours do not depend on the density $\rho$ of the particles, but only on $\xi$. This is not expected to hold for a step of density $\rho_+ \neq \rho_-$.

The knowledge of the statistical properties of $J_t$ allows to deduce those of the position of a tracer $X_t$ [3,4]. For instance, we recover the known variance [12],

$$\lim_{t\to\infty} \frac{1}{\sqrt{t}} \left\langle X_t^2 \right\rangle = \frac{2(1-\rho)}{\rho\sqrt{\pi}}, \tag{24}$$

and obtain the covariance

$$\lim_{t\to\infty} \frac{1}{\sqrt{t}} \left\langle Q_t X_t \right\rangle = \frac{2(1-\rho)}{\sqrt{\pi}}, \tag{25}$$

from which we deduce that these quantities are fully correlated, since

$$\lim_{t\to\infty} \frac{\left\langle Q_t X_t \right\rangle}{\sqrt{\left\langle Q_t^2 \right\rangle \left\langle X_t^2 \right\rangle}} = 1. \tag{26}$$

This is due to the fact that, $Q_t = \rho X_t$ at leading order in time, which results from

$$\lim_{t\to\infty} \frac{1}{\sqrt{t}} \left\langle (Q_t - \rho X_t)^2 \right\rangle = 0. \tag{27}$$

However, the two observables $Q_t$ and $X_t$ are not simply proportional, even in the long time limit, since for instance

$$\lim_{t\to\infty} \frac{1}{\sqrt{t}} \ln \left\langle e^{\lambda(Q_t - \rho X_t)} \right\rangle = \frac{1}{4\sqrt{\pi}} \rho(1-\rho)^3 \lambda^4 + \mathcal{O}(\lambda^5). \tag{28}$$

This means that the variance of $Q_t - \rho X_t$ is nonzero, since the higher order cumulants do not vanish. However, the expression of this variance is out of reach of our approach, since we can only describe the leading $\sqrt{t}$ with the MFT. Note that (28) is not symmetric in $\rho$ and $1-\rho$, since following a tracer (which is a particle) breaks the particle-hole symmetry of the SEP. In particular, Eq. (28) vanishes faster when $\rho \to 1$ compared to $\rho \to 0$. In the dense limit, both $\ln\left\langle e^{\lambda Q_t} \right\rangle$ and $\ln\left\langle e^{\chi X_t} \right\rangle$ are of order $1-\rho$, while (28) vanishes as $(1-\rho)^3$. This shows that $Q_t = X_t$ in this limit, as can be seen from the fact that they have identical cumulants in that case [9, 19]. On the other hand, in the dilute limit, $X_t = \mathcal{O}(1/\rho)$ and $Q_t = \mathcal{O}(1)$, with $\ln\left\langle e^{\lambda Q_t} \right\rangle = \mathcal{O}(\rho)$. This time (28) does not vanish faster than the cumulants of the current, and thus $Q_t \neq \rho X_t$, as illustrated from the fact that their cumulants differ [9,17] in that case.

In addition to the joint statistical properties of $Q_t$ and $J_t$, we also obtain the profiles $\Phi$ and $\bar{\Phi}$ which quantify the correlation between the density of particles and the observables, at final and initial times. At lowest orders in $\lambda$ and $\nu$, we recover the profiles obtained previously [6,7]

$$\langle \eta_r Q_t \rangle_c \underset{t\to\infty}{\simeq} \text{sign}(x) \frac{\rho(1-\rho)}{2} \text{erfc}\left(\frac{|x|}{2}\right), \quad \langle \eta_r J_t \rangle_c \underset{t\to\infty}{\simeq} \frac{\rho(1-\rho)}{2} \begin{cases} \text{erfc}\left(\frac{x}{2}\right), & \text{for } x > \xi, \\ -\text{erfc}\left(-\frac{x}{2}\right), & \text{for } x < \xi, \end{cases} \tag{29}$$

with $x = r/\sqrt{t}$. We additionally obtain the joint profiles, such as

$$\langle \eta_r Q_t J_t \rangle_c \underset{t\to\infty}{\simeq} \frac{\rho(1-\rho)(1-2\rho)}{2} \begin{cases} \text{erfc}\left(\frac{x}{2}\right), & \text{for } x > \xi, \\ 0, & \text{for } 0 < x < \xi, \\ \text{erfc}\left(-\frac{x}{2}\right), & \text{for } x < 0. \end{cases} \tag{30}$$

To understand the meaning of this expression, we can rewrite it as

$$\langle \eta_r Q_t J_t \rangle_c = \text{Cov}\left(\eta_r, \widetilde{\text{cov}}(Q_t, J_t)\right), \quad \text{where} \quad \widetilde{\text{cov}}(Q_t, J_t) = (Q_t - \langle Q_t \rangle)(J_t - \langle J_t \rangle), \tag{31}$$

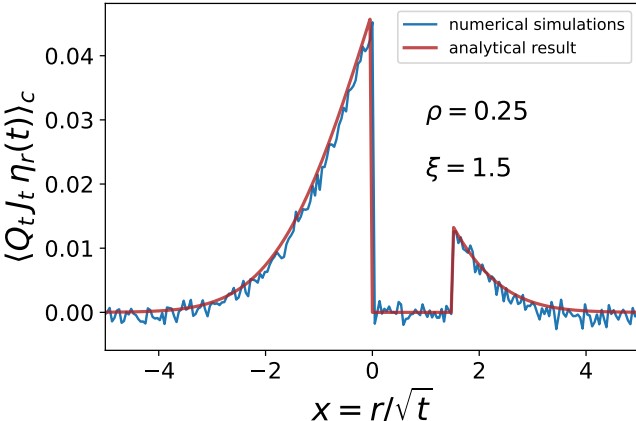

Figure 2: Correlation function $\langle \eta_r Q_t J_t \rangle$ as a function of $r/\sqrt{t}$, for a mean density of the SEP $\rho = 0.25$, and for $\xi = 1.5$. The solid red line is the analytical result (30). The blue line is the result of a direct numerical simulation of the SEP with 1000 sites, 250 particles, measured at time $t = 1000$ and averaged over $10^7$ realisations.

is the empirical covariance. This means that this profile measures the covariance between the density of particles on one hand, and the correlations between $Q_t$ and $J_t$ on the other hand. From (30), we see that it is positive for $\rho < \frac{1}{2}$ and negative for $\rho > \frac{1}{2}$. This means that adding particles when $\rho < \frac{1}{2}$ increases the correlation between $Q_t$ and $J_t$, while it decreases it when $\rho > \frac{1}{2}$. This behavior is expected, since the maximal currents are reached for $\rho = \frac{1}{2}$. In addition, (30) is extremal near $x = 0^-$ and $x = \xi^+$. In other words, a change of the number of particles in these sectors affects strongly the correlation between $Q_t$ and $J_t$. This is due to the fact that the particles in these regions are more likely than distant particles to cross the two "walls" $x = 0$ and $x = \xi$ and thus affect both $Q_t$ and $J_t$. Conversely, the profile (30) vanishes between 0 and $\xi$, indicating that adding more particles in that region does not affect the correlation between $Q_t$ and $J_t$. This is also expected, since these particles can only cross one "wall"[2] either at $x = 0$ or $x = \xi$, and can thus affect only one of these observables. This is confirmed by numerical simulations of the SEP, as shown in Fig. 2.

Beyond the perturbative expansion in $\lambda$ and $\nu$, we can also plot the profiles for finite values of the these parameters by solving numerically the integral equations (8,9). These profiles have the interpretation of mean density profiles under the condition that $Q_t$ and $J_t$ take given values [6]. They are represented in Fig. 1. For instance, the plot in Fig. 1 (left) corresponds to having currents $Q_t$ and $J_t$ larger than their mean values. This is why there is an accumulation of particles to the right of the two "walls" $x = 0$ and $x = \xi$, and a depletion to the left. Conversely, for the corresponding initial profile $\bar{\Phi}$ (Fig. 1, right), there is an increase of particles to the left of the origin, and a depletion to the right, so that with the diffusive time evolution, these particles will cross the origin, and $\xi$, to contribute to a larger value of $Q_t$ and $J_t$.

## 2.3 Discussion

*Integral equations.* — The Wiener-Hopf integral equation obtained previously in the case of a single observable ($Q_t$, $J_t$ or $X_t$) [6,7] can be recovered from (8) and (9) by setting either $\lambda = 0$ or $\nu = 0$. This corresponds to $\alpha = 0$ or $\beta = 0$ respectively in Eqs (8,9), so that the two profiles at initial and final time satisfy the same equation, as noted previously [7,8].

---

[2]Particles can of course cross several times both walls, but an odd number of times only one of them, resulting in this net effect.

Note that if we set $\xi = 0$, we recover the equations of [6,7]. This is expected since then, $J_t = Q_t$ and thus $\Phi$ and $\bar{\Phi}$ involve a single observable.

In the case $\alpha \neq 0$ and $\beta \neq 0$, the equation for the profile at final time (8) is more complicated than the one obtained in the case of a single observable [6,7], as it now involves a double convolution. On the other hand, the equation for the initial profile (9) keeps a simpler Wiener-Hopf structure, but with a more complicated kernel which involves the solution at final time (10).

These integral equations extend the ones discovered in [6,7] in the case of a single observable (current $Q_t$ or $J_t$, tracer position $X_t$). This further emphasises the key role of such strikingly simple integral equations involving partial convolutions in interacting particle systems.

*Boundary conditions.* — We stress that the boundary equations (11,12) hold for any single-file system, and not only for the SEP. Furthermore, these equations take a simple physical form in terms of the chemical potential $\mu$, which can be written as

$$\mu(\rho) = \int^{\rho} \frac{2D(r)}{\sigma(r)} \mathrm{d}r \,, \tag{32}$$

in terms of the diffusion coefficient $D$ and the mobility $\sigma$, which describe the system at large scale [26–28]. For the SEP, $D(\rho) = 1$ and $\sigma(\rho) = 2\rho(1-\rho)$, which gives the expression (14) for the chemical potential of the SEP.

Explicitly, (11) states that the chemical potential in the system is discontinuous at $x = 0$ and $x = \xi$, with a discontinuity given by the parameters $\lambda$ and $\nu$ of the joint generating function (4). From a physical point of view, this can be understood as follows. The parameters $\lambda$ and $\nu$ play the role of conjugate variables (in the sense of thermodynamics) to the integrated currents $Q_t$ and $J_t$, which count particles. The conjugate quantity to the particle number being the chemical potential, it is expected that $\lambda$ and $\nu$ are related to the chemical potential. Finally, $\lambda$ and $\nu$ have an effect on the density of particles. For instance, when $\lambda > 0$, the exponentials in the definitions of the profiles (5) give more weight to the realisations in which $Q_t > 0$, and thus we expect an increase of the number of particles to the right of the origin. This results from a higher chemical potential to the right of the origin compared to the left, as described by (11).

Remarkably, the equations (11,12) obtained for the case of the currents $Q_t$ and $J_t$ can be extended to other observables. For instance, it has been recently shown that the current $Q_t$ in a single-file model can be mapped onto the position of a tracer in a dual single-file model [34]. Under this mapping, the relations (11,12) become, for the new system (see Appendix A)

$$P(\Phi(0^+)) - P(\Phi(0^-)) = \lambda \,, \quad \partial_x \mu(\Phi)|_{0^+} = \partial_x \mu(\Phi)|_{0^-} \,, \tag{33}$$

where $P$ is the pressure[3]

$$P(\rho) = \int^{\rho} \frac{2rD(r)}{\sigma(r)} \mathrm{d}r \,, \tag{34}$$

and $\Phi$ is now the long time limit of the correlation between the position $X_t$ of the tracer and the density in the reference frame of the tracer, $\Phi(x = r/\sqrt{t}) \underset{t\to\infty}{\simeq} \left\langle \rho(r + X_t, t)e^{\lambda X_t} \right\rangle / \left\langle e^{\lambda X_t} \right\rangle$.

Finally, in the case of the SEP, these relations (11,12) and (33) reduce to the ones obtained from microscopic considerations in [5–7]. The precise relation with the corresponding microscopic equations is given in Appendix B.

---

[3]The relation involving the pressure Eq.(33), left, has been first guessed by Alexis Poncet during private exchanges prior to this work.

# 3 Hydrodynamic description using macroscopic fluctuation theory

The Macroscopic Fluctuation Theory (MFT) gives an effective description at large scales of a diffusive system [26–28]. Introducing a scaling factor $T$, which corresponds to the large observation time, the density of particles is defined as

$$\rho_T(x,t) = \frac{1}{\sqrt{T}} \sum_i \eta_i(tT) \delta\left(x - \frac{i}{\sqrt{T}}\right). \tag{35}$$

In the limit $T \to \infty$, this density converges to a continuous stochastic function $\rho(x,t)$. The probability of observing an initial profile $\rho(x,0)$ evolves to another profile $\rho(x,1)$ at time $t = 1$ (corresponding to the large time $T$ in the SEP) takes a large deviation form [10]

$$\mathbb{P}[\rho(x,0) \to \rho(x,t)] \simeq \int \mathcal{D}[\rho(x,t)]\mathcal{D}[H(x,t)]e^{-\sqrt{T}\mathcal{S}[\rho,H]}, \tag{36}$$

where $H$ is a conjugate field, and $S$ is the MFT action

$$\mathcal{S}[\rho,H] = \int_{-\infty}^{\infty} dx \int_0^1 dt \left[ H\partial_t\rho + D(\rho)\partial_x\rho\,\partial_xH - \frac{\sigma(\rho)}{2}(\partial_xH)^2 \right]. \tag{37}$$

For the SEP, $D(\rho) = 1$ and $\sigma(\rho) = 2\rho(1-\rho)$. This result was first proved for the SEP [35], and was later extended to arbitrary 1D diffusive systems, which can be described by other transport coefficients $D(\rho)$ and $\sigma(\rho)$ [26–28]. See for instance [34] for a list of models, and their corresponding coefficients.

Initially, the system is described by the random density $\rho(x,0)$, which fluctuates around a given density $\rho_0(x)$, of distribution [10]

$$\mathbb{P}[\rho(x,0)] \simeq e^{-\sqrt{T}\mathcal{F}[\rho(x,0)]}, \quad \mathcal{F}[\rho(x,0)] = \int dx \int_{\rho_0(x)}^{\rho(x,0)} dz \frac{2D(z)}{\sigma(z)}(\rho(x,0)-z). \tag{38}$$

For the moment, we consider an arbitrary initial condition $\rho_0(x)$, which we will later specify to the case $\rho_0(x) = \rho_+\Theta(x)+\rho_-\Theta(-x)$. Microscopically, it corresponds to picking independently, for each site $i$ of the SEP, an occupation number $\eta_i(0) = 1$ with probability $\rho_0(i/\sqrt{T})$. In the continuous limit, this becomes (38).

The two currents $Q_t$ (1) and $J_t$ (3) can be expressed in terms of the density $\rho(x,t)$ (35) as

$$\frac{Q_T}{\sqrt{T}} \equiv \mathcal{Q}[\rho] = \int_0^{\infty} [\rho(x,1)-\rho(x,0)]dx, \tag{39a}$$

$$\frac{J_T}{\sqrt{T}} \equiv \mathcal{J}[\rho] = -\rho_+\xi + \int_{-\infty}^{\infty} [(\rho(x,1)-\rho_+)\Theta(x-\xi)-(\rho(x,0)-\rho_+)\Theta(x)]dx. \tag{39b}$$

Within this formalism, the joint moment generating function of $Q_T$ and $J_T$ reads

$$\left\langle e^{\lambda Q_T + \nu J_T} \right\rangle = \int \mathcal{D}[\rho(x,t)]\mathcal{D}[H(x,t)] \int \mathcal{D}[\rho(x,0)]e^{-\sqrt{T}(\mathcal{S}[\rho,H]+\mathcal{F}[\rho(x,0)]-\lambda\mathcal{Q}[\rho]-\nu\mathcal{J}[\rho])}. \tag{40}$$

In the long time limit $T \to \infty$, these integrals can be evaluated by a saddle point method, which yields

$$\hat{\psi}_\xi(\lambda,\nu) = \lim_{T\to\infty} \frac{1}{\sqrt{T}} \ln\left\langle e^{\lambda Q_t + \nu J_t} \right\rangle = \lambda\mathcal{Q}[q] + \nu\mathcal{J}[q] - \mathcal{S}[q,p] - \mathcal{F}[q(x,0)], \tag{41}$$

where we have denoted $(q, p)$ the saddle point of $(\rho, H)$. It can be determined by minimising the terms in the exponential, which yields the MFT equations

$$\partial_t q = \partial_x[D(q)\partial_x q] - \partial_x[\sigma(q)\partial_x p], \tag{42}$$

$$\partial_t p = -D(q)\partial_x^2 p - \frac{1}{2}\sigma'(q)(\partial_x p)^2, \tag{43}$$

completed by the boundary conditions

$$p(x, 1) = \lambda\Theta(x) + \nu\Theta(x - \xi), \quad p(x, 0) = (\lambda + \nu)\Theta(x) + \int_{\rho_0(x)}^{q(x,0)} dr \frac{2D(r)}{\sigma(r)}. \tag{44}$$

The expression of the cumulant generating function can be simplified by taking a derivative with respect to $\lambda$ or $\nu$, and using that the saddle point solution $(q, p)$ is the minimum of the action:

$$\partial_\lambda \hat{\psi}_\xi = \mathcal{Q}[q], \quad \partial_\nu \hat{\psi}_\xi = \mathcal{J}[q]. \tag{45}$$

These are standard relations in the context of large deviations [36, 37].

The profiles $w_r$ (5) can be obtained from the same procedure, for instance,

$$w_r(\lambda, \nu, T)$$
$$\simeq \frac{\int \mathcal{D}[\rho(x,t)]\mathcal{D}[H(x,t)] \int \mathcal{D}[\rho(x,0)]\rho\left(\frac{r}{\sqrt{T}}, 1\right) e^{-\sqrt{T}(\mathcal{S}[\rho,H]+\mathcal{F}[\rho(x,0)]-\lambda\mathcal{Q}[\rho]-\nu\mathcal{J}[\rho])}}{\int \mathcal{D}[\rho(x,t)]\mathcal{D}[H(x,t)] \int \mathcal{D}[\rho(x,0)] e^{-\sqrt{T}(\mathcal{S}[\rho,H]+\mathcal{F}[\rho(x,0)]-\lambda\mathcal{Q}[\rho]-\nu\mathcal{J}[\rho])}}. \tag{46}$$

Performing again the saddle point estimate, we obtain,

$$w_r(\lambda, \nu, T) \underset{T\to\infty}{\simeq} q\left(x = \frac{r}{\sqrt{T}}, 1\right) \equiv \Phi(x). \tag{47}$$

Similarly, the correlation with the initial occupations $\bar{w}_r$ (6) reads

$$\bar{w}_r(\lambda, \nu, T) \underset{T\to\infty}{\simeq} q\left(x = \frac{r}{\sqrt{T}}, 0\right) \equiv \bar{\Phi}(x). \tag{48}$$

The MFT profile $q$ at initial and final time actually coincides with the correlations $w_r$ and $\bar{w}_r$ in the long time limit, as shown in [5, 6]. Furthermore, the joint cumulant generating function $\hat{\psi}_\xi$ is fully determined by the knowledge of the profile $q$, thanks to (45). Our goal is thus to determine these profiles.

## 4 The example of the low density limit

The MFT equations for the SEP (42, 43) being rather complicated, we first focus on the simpler case of the low density limit. In this limit, the SEP becomes equivalent to a model of reflecting Brownian particles on the real line [3]. The MFT equations reduce to

$$\partial_t q = \partial_x[\partial_x q] - \partial_x[2q\partial_x p], \tag{49}$$

$$\partial_t p = -\partial_x^2 p - (\partial_x p)^2. \tag{50}$$

These equations can be reduced to diffusion equations by the Cole-Hopf transform $P = e^p$ and $Q = qe^{-p}$ [10, 21], so that

$$\partial_t Q = \partial_x^2 Q, \quad \partial_t P = -\partial_x^2 P. \tag{51}$$

The initial and final conditions (44) become

$$P(x, 1) = e^{\lambda\Theta(x) + \nu\Theta(x-\xi)}, \quad Q(x, 0) = \rho_0(x)e^{-(\lambda+\nu)\Theta(x)}. \tag{52}$$

We straightforwardly obtain the solution $q = QP$, and thus the profiles both at initial and final times,

$$\Phi(x) = q(x, 1) = e^{\lambda\Theta(x) + \nu\Theta(x-\xi)} \int_{-\infty}^{\infty} dz\, \rho_0(z) e^{-(\lambda+\nu)\Theta(z)} \frac{e^{-(x-z)^2/4}}{\sqrt{4\pi}}, \tag{53}$$

$$\bar{\Phi}(x) = q(x, 0) = \rho_0(x)e^{-(\lambda+\nu)\Theta(x)}\left[e^{\lambda+\nu} - \frac{e^{\lambda}-1}{2}\operatorname{erfc}\left(\frac{x}{2}\right) - e^{\lambda}\frac{e^{\nu}-1}{2}\operatorname{erfc}\left(\frac{x-\xi}{2}\right)\right], \tag{54}$$

where we have assumed that $\xi > 0$ for $\bar{\Phi}$. The expression is similar in the case $\xi < 0$. The cumulant generating function can be obtained from the relations (45), which gives

$$\partial_\lambda \hat{\psi}_\xi = \int_{-\infty}^{\infty} dx\, \rho_0(x)e^{-(\lambda+\nu)\Theta(x)} \int_{-\infty}^{\infty} dz\, e^{\lambda\Theta(z)+\nu\Theta(z-\xi)} \frac{e^{-(x-z)^2/4}}{\sqrt{4\pi}}[\Theta(z) - \Theta(x)], \tag{55}$$

$$\partial_\nu \hat{\psi}_\xi = \int_{-\infty}^{\infty} dx\, \rho_0(x)e^{-(\lambda+\nu)\Theta(x)} \int_{-\infty}^{\infty} dz\, e^{\lambda\Theta(z)+\nu\Theta(z-\xi)} \frac{e^{-(x-z)^2/4}}{\sqrt{4\pi}}[\Theta(z-\xi) - \Theta(x)]. \tag{56}$$

Integrating with the initial value $\hat{\psi}(0, 0) = 0$, we get

$$\hat{\psi}_\xi(\lambda, \nu) = \int_{-\infty}^{\infty} dx\, \rho_0(x) \int_{-\infty}^{\infty} dz\left[e^{\lambda\Theta(z)+\nu\Theta(z-\xi)-(\lambda+\nu)\Theta(x)} - 1\right]\frac{e^{-(x-z)^2/4}}{\sqrt{4\pi}}. \tag{57}$$

Note that this expression is compatible with the very recent study [38].

Finally, the density profiles of the SEP assume simple explicit forms in the low density limit.

## 5   Derivation of the main equations

We now address the case of arbitrary density of the SEP. To obtain the equations satisfied by the initial and final profiles $\bar{\Phi}$ and $\Phi$, we will rely on the inverse scattering approach which has recently been applied to solve systems of equations related to (42,43), in the context of the KPZ equation or MFT [8, 22–25, 39]. As we will see below, this formalism is powerful to obtain the bulk equations for $\Phi$ and $\bar{\Phi}$, but introduces unknown constants which can be tricky to determine. Here, we will obtain these constants by making use of boundary conditions which are deduced from the MFT equations (42-44).

### 5.1   Boundary conditions

We first derive the boundary conditions (11-13), which are direct consequence of the MFT equations (42-44). These equations will take a simple form, in terms of physical quantities. The equation satisfied by $p$ (43) is an antidiffusion, with no singularity in the r.h.s. for $t < 1$, except a discontinuity for $q$ at $t = 0$. Therefore, the solution $p(x, 0)$ is a smooth function of $x$, and in particular at $x = 0$. The boundary conditions are thus straightforwardly deduced from the initial condition (44), which takes the from,

$$p(x, 0) = (\lambda + \nu)\Theta(x) + \mu(q(x, 0)) - \mu(\rho_0(x)), \tag{58}$$

where we have introduced the chemical potential $\mu(\rho)$, defined by (32). Evaluating (58) at $x = 0^+$ and $x = 0^-$, and taking the difference and using the continuity of $p(x,0)$ at $x = 0$, we get the first boundary condition for $q(x,0) \equiv \bar{\Phi}(x)$

$$\mu(\bar{\Phi}(0^+)) - \mu(\bar{\Phi}(0^-)) = -(\lambda + \nu) + \mu(\rho_0(0^+)) - \mu(\rho_0(0^-)). \tag{59}$$

Similarly, writing the continuity of the first derivative of $p(x,0)$ at $x = 0$, we deduce from (58)

$$\partial_x \mu(\bar{\Phi})\big|_{0^+} - \partial_x \mu(\bar{\Phi})\big|_{0^-} = \partial_x \mu(\rho_0)|_{0^+} - \partial_x \mu(\rho_0)|_{0^-}. \tag{60}$$

For $\rho_0(x) = \rho_+ \Theta(x) + \rho_- \Theta(-x)$, these relations become (13).

   To obtain the conditions at final time, we rely on a time-reversal mapping, which extends the time-reversal symmetry discussed in [10] for the case of the current $Q_t$. In that work, the MFT action was found to be invariant under time reversal symmetry $\rho(x,t) \to \rho(x,1-t)$ and $j(x,t) \to -j(x,1-t)$, with a density $\rho$ and a current $j$ satisfying the conservation relation $\partial_t \rho + \partial_x j = 0$. At the saddle point of the MFT action, the density becomes $q$ and the current becomes $j = -D(q)\partial_x q + \sigma(q)\partial_x p$ [10]. The time reversal symmetry then becomes $q(x,t) \to q(x,1-t)$, $j(x,t) = [-D(q)\partial_x q + \sigma(q)\partial_x p]|_{(x,t)} \to -[-D(q)\partial_x q + \sigma(q)\partial_x p]|_{(x,1-t)}$. Here, we do not have the time-reversal symmetry, because at final time the currents are measured at positions 0 and $\xi$, which are different from the initial position 0. Nevertheless, we define two new fields $\hat{q}$ and $\hat{p}$ by

$$q(x,t) = \hat{q}(x,1-t), \quad \partial_x p(x,t) = -\partial_x \hat{p}(x,1-t) + \frac{2D(\hat{q})}{\sigma(\hat{q})}\partial_x \hat{q}\bigg|_{(x,1-t)}. \tag{61}$$

Integrating the second relation gives

$$p(x,t) = -\hat{p}(x,1-t) + \mu(\hat{q}(x,1-t)) + c, \tag{62}$$

with $c$ a constant. Inserting these relations into the MFT equations (42,43), we find that $\hat{q}$ and $\hat{p}$ obey the same equations:

$$\partial_t \hat{q} = \partial_x [D(\hat{q})\partial_x \hat{q}] - \partial_x [\sigma(\hat{q})\partial_x \hat{p}], \quad \partial_t \hat{p} = -D(\hat{q})\partial_x^2 \hat{p} - \frac{1}{2}\sigma'(\hat{q})(\partial_x \hat{p})^2. \tag{63}$$

This was already noticed in [40]. The initial and final conditions (44) become

$$\hat{p}(x,0) = \mu(\hat{q}(x,0)) + c - \lambda\Theta(x) - \nu\Theta(x-\xi), \quad \hat{p}(x,1) = -(\lambda + \nu)\Theta(x) + \mu(\rho_0(x)) - c. \tag{64}$$

These conditions are different from the original ones (44), and they are the source of the breaking of time-reversal symmetry.[4] We can however still use a similar argument as we used above at $t = 0$. The conjugate field $\hat{p}$ obeys an antidiffusion equation, which is not singular for $t < 1$. Therefore $\hat{p}(x,0)$ is smooth. From (64) left, this straightforwardly yields the conditions for $\hat{q}(x,0) = q(x,1) \equiv \Phi(x)$,

$$\mu(\Phi(0^+)) - \mu(\Phi(0^-)) = \lambda, \quad \mu(\Phi(\xi^+)) - \mu(\Phi(\xi^-)) = \nu, \tag{65}$$

and from the derivative,

$$\partial_x \mu(\Phi)\big|_{0^+} = \partial_x \mu(\bar{\Phi})\big|_{0^-}, \quad \partial_x \mu(\Phi)\big|_{\xi^+} = \partial_x \mu(\bar{\Phi})\big|_{\xi^-}. \tag{66}$$

---

[4]If we consider the current $Q_t$ only, $\nu = 0$. In the case of an initial step density $\rho_0(x) = \rho_+ \Theta(x) + \rho_- \Theta(-x)$, the system has time-reversal symmetry. Indeed, by choosing $c = \mu(\rho_-)$, the new initial and final conditions are identical to the original ones, upon changing $\lambda \to \mu(\rho_+) - \mu(\rho_-) - \lambda$. This is indeed the relation found in [10].

These are the relations (11,12) announced above. Note that these equations for $\Phi$ hold for *any* initial density profile $\rho_0$.

   *Important remark:* We have derived the boundary conditions for $\Phi$ in the case of an annealed initial condition. One could also consider a quenched initial condition, which corresponds to $q(x,0) = \rho_0(x)$. In this case, the mapping (61) can still be performed. One obtains the same MFT equations (42,43), but with the initial and final conditions

$$\hat{p}(x,0) = \mu(\hat{q}(x,0)) + c - \lambda\Theta(x) - \nu\Theta(x-\xi), \quad \hat{p}(x,1) = -p(x,0) + \mu(\rho_0(x)) - c. \quad (67)$$

The second relation involves the unknown function $p(x,0)$, which is smooth, but the first relation is identical to the annealed case. The same argument as above applies, and the boundary conditions (65,66) still hold in the quenched case.

## 5.2 Bulk equations

### 5.2.1 Mapping to the AKNS equations

We adapt the inverse scattering approach that was applied to the case of the integrated current $Q_t$ in the SEP in [8] to the case of the joint distribution of $Q_t$ and $J_t$. The first step is to introduce the new functions [8]

$$u = \frac{1}{(1-2q)}\partial_x\left[q(1-q)e^{-\int_{-\infty}^x(1-2q)\partial_x p}\right], \quad v = -\frac{1}{1-2q}\partial_x e^{\int_{-\infty}^x(1-2q)\partial_x p}. \quad (68)$$

Under this transformation, the MFT equations for the SEP (42,43) become the AKNS equations [41]

$$\partial_t u = \partial_x^2 u - 2u^2 v, \quad \partial_t v = -\partial_x^2 v + 2uv^2. \quad (69)$$

These equations are integrable and can be solved using the inverse scattering transform [29]. Before entering the resolution in more details, let us study the initial and final conditions for $u$ and $v$. From the conditions on $p$ and $q$ (44) and the transformation (68), we obtain

$$
\begin{aligned}
u(x,0) &= \left[\partial_x q - q(1-q)\partial_x p\right]e^{-\int_{-\infty}^x(1-2q)\partial_x p}\Big|_{t=0} \\
&= q(1-q)\left[(\lambda+\nu)\delta(x) - \frac{\partial_x\rho_0}{\rho_0(1-\rho_0)}\right]e^{-\int_{-\infty}^x(1-2q)\partial_x p}\Big|_{t=0}.
\end{aligned}
\quad (70)
$$

From now on, we consider the case of a step initial density $\rho_0(x) = \rho_+\Theta(x) + \rho_-\Theta(-x)$. In Eq. (70), the term $\partial_x\rho_0$ thus gives another $\delta(x)$ term, but with an unknown prefactor, because $\rho_0$ is discontinuous at 0. Even in the case of a constant density $\rho_+ = \rho_-$, the prefactor of the remaining $\delta$ function is unknown, because $q$ is discontinuous at $x = 0$. Therefore, we can only write

$$u(x,0) = c_0\,\delta(x), \quad (71)$$

with an unknown constant $c_0$. Similarly, for $v(x, t = 1)$ at final time, we get

$$v(x,1) = -\partial_x p\, e^{\int_{-\infty}^x(1-2q)\partial_x p}\Big|_{t=1} = c_1\,\delta(x) + c_2\,\delta(x-\xi), \quad (72)$$

with two other unknown constants $c_1$ and $c_2$, coming from the fact that the term in the exponential is not well defined since $p$ and $q$ are discontinuous at $x = 0$ and $x = \xi$. Actually, we can get rid of one of these unknown constants by using the invariance of the AKNS equations (69) under the transformation $u(x,t) \to u(x,t)/K$ and $v(x,t) \to Kv(x,t)$. Choosing $K = c_0$, we have the initial and final conditions

$$u(x,0) = \delta(x), \quad v(x,1) = \alpha\,\delta(x) + \beta\,\delta(x-\xi), \quad (73)$$

with $\alpha = c_1 c_0$ and $\beta = c_2 c_0$. We will see below how we can determine the constants $\alpha$ and $\beta$. The simplicity of these conditions will allow for an explicit solution of the AKNS equations at initial and final times. Furthermore, this solution will yield the desired equations for the profiles, since $\partial_x p(x, 1)$ is a sum of $\delta$ functions as seen from (44),

$$u(x, 1) = \frac{1}{K} \left[ \partial_x q - q(1-q)\partial_x p \right] e^{-\int_{-\infty}^{x}(1-2q)\partial_x p} \bigg|_{t=1} = \begin{cases} a_- \partial_x q(x, 1), & \text{for } x < 0, \\ a_0 \, \partial_x q(x, 1), & \text{for } 0 < x < \xi, \\ a_+ \partial_x q(x, 1), & \text{for } x > \xi, \end{cases} \tag{74}$$

with different proportionality constants $a_-$, $a_0$ and $a_+$ for each domain $x < 0$, $0 < x < \xi$ and $x > \xi$ because $p(x, 1)$ is discontinuous at both $x = 0$ and $x = \xi$, hence the value of the exponential differs in each interval (by an unknown factor, since $q(x, 1)$ is also discontinuous at these points). Similarly,

$$v(x, 0) = -K \partial_x p \, e^{\int_{-\infty}^{x}(1-2q)\partial_x p} \bigg|_{t=0} . \tag{75}$$

We can simplify this equation in the following way. We take the derivative of the boundary condition (44) at $t = 0$, which gives,

$$\partial_x p(x, 0) = \frac{\partial_x q(x, 0)}{q(1-q)} + c_3 \delta(x) \quad \Rightarrow \quad \int_{-\infty}^{x}(1-2q)\partial_x p \bigg|_{t=0} = \ln \frac{q(1-q)}{\rho_-(1-\rho_-)} + c_4 \Theta(x), \tag{76}$$

with new constants $c_3$ and $c_4$ (which we will not need to determine). Indeed, combining with (75), we get

$$v(x, 0) = \begin{cases} b_- \partial_x q(x, 0), & \text{for } x < 0, \\ b_+ \partial_x q(x, 0), & \text{for } x > 0, \end{cases} \tag{77}$$

with two different proportionality constants $b_-$ and $b_+$ for $x < 0$ and $x > 0$.

To summarize, the solutions $u(x, t)$ and $v(x, t)$ of the AKNS equations are directly related to the derivative of the profiles at initial (48) and final times (47),

$$\Omega(x) \equiv u(x, 1) = \begin{cases} a_- \Phi'(x), & \text{for } x < 0, \\ a_0 \, \Phi'(x), & \text{for } 0 < x < \xi, \\ a_+ \Phi'(x), & \text{for } x > \xi, \end{cases} \quad \bar{\Omega}(x) \equiv v(x, 0) = \begin{cases} b_- \bar{\Phi}'(x), & \text{for } x < 0, \\ b_+ \bar{\Phi}'(x), & \text{for } x > 0, \end{cases} \tag{78}$$

with constants $a_-$, $a_0$, $a_+$, $b_-$ and $b_+$ that will be determined by the boundary conditions derived in Section 5.1. The other relations are given by (73), with the constants $\alpha$ and $\beta$ that remain to be determined.

### 5.2.2 Solution using the scattering technique

Our goal is now to obtain integral equations verified by $\Omega$ and $\bar{\Omega}$. To solve the AKNS equations we rely on the standard approach [29], recently used in [8,22,24], and introduce the auxiliary linear problem for the two-component vector $\Psi$,

$$\partial_x \Psi = U\Psi, \quad \partial_t \Psi = V\Psi, \quad U = \begin{pmatrix} -ik & v \\ u & ik \end{pmatrix}, \quad V = \begin{pmatrix} 2k^2 + uv & 2ikv - \partial_x v \\ 2iku + \partial_x u & -2k^2 - uv \end{pmatrix}. \tag{79}$$

The compatibility condition between the first two equations, $\partial_x \partial_t \Psi = \partial_t \partial_x \Psi$ is equivalent to the AKNS equations (69). The idea is therefore to solve the simpler linear problem (79), and deduce the solution for $u(x, t)$ and $v(x, t)$. Since $\partial_x q \to 0$ and $\partial_x p \to 0$ for $x \to \pm\infty$, $u(x, t)$ and $v(x, t)$ decay to 0 at $\pm\infty$. The matrix $U$ then becomes diagonal at $\pm\infty$, and therefore $\Psi$

is a superposition of plane waves in this limit. We introduce two independent solutions $\phi$ and $\bar{\phi}$, defined by their behaviour at $-\infty$,

$$\phi(x,t) \underset{x\to-\infty}{\simeq} e^{2k^2t}\begin{pmatrix} e^{-ikx} \\ 0 \end{pmatrix}, \quad \bar{\phi}(x,t) \underset{x\to-\infty}{\simeq} e^{-2k^2t}\begin{pmatrix} 0 \\ -e^{ikx} \end{pmatrix}, \tag{80}$$

where we have placed the factors $e^{\pm 2k^2t}$ so that $\phi$ and $\bar{\phi}$ satisfy the time evolution equation (79) at $-\infty$. For $x \to +\infty$, we can write the solution as the superposition of the same two plane waves,

$$\phi(x,t) \underset{x\to+\infty}{\simeq} \begin{pmatrix} a(k,t)e^{-ikx} \\ b(k,t)e^{ikx} \end{pmatrix}, \quad \bar{\phi}(x,t) \underset{x\to+\infty}{\simeq} \begin{pmatrix} \bar{b}(k,t)e^{-ikx} \\ -\bar{a}(k,t)e^{ikx} \end{pmatrix}. \tag{81}$$

This defines a *scattering problem*, in which plane waves at $-\infty$ are scattered by the *potentials* $u(x,t)$ and $v(x,t)$ into a superposition of plane waves at $+\infty$. The coefficients $a$, $\bar{a}$, $b$, $\bar{b}$ are called the *scattering amplitudes*. All the information on the functions $u(x,t)$ and $v(x,t)$ are encoded in the scattering amplitudes, so that $u(x,t)$ and $v(x,t)$ can be reconstructed from $a$, $\bar{a}$, $b$, $\bar{b}$. This is called the *inverse scattering* procedure, and is quite complicated to do in practice. Here, we will follow a different route, used in [8, 22–25, 39, 42]: we will determine the scattering amplitudes at initial and final times in terms of the functions $u(x,1) = \Omega(x)$ and $v(x,0) = \bar{\Omega}(x)$, and relate them using the time evolution (79) to obtain integral equations satisfied by these functions. Indeed, one strength of the scattering approach is that it transforms the complicated time evolution of the AKNS equations (69) into a very simple time dependence of the scattering amplitudes. Their time evolution can be computed using the matrix $V(+\infty) = \text{Diag}(2k^2, -2k^2)$ in (79) at $+\infty$, which directly gives

$$\partial_t a(k,t) = 2k^2 a(k,t), \qquad \qquad \partial_t b(k,t) = -2k^2 b(k,t), \tag{82a}$$

$$\partial_t \bar{a}(k,t) = -2k^2 \bar{a}(k,t), \qquad \qquad \partial_t \bar{b}(k,t) = 2k^2 \bar{b}(k,t), \tag{82b}$$

and thus

$$a(k,t) = e^{2k^2t} a(k,0), \qquad \qquad b(k,t) = e^{-2k^2t} b(k,0), \tag{83a}$$

$$\bar{a}(k,t) = e^{-2k^2t} \bar{a}(k,0), \qquad \qquad \bar{b}(k,t) = e^{2k^2t} \bar{b}(k,0). \tag{83b}$$

There only remains to determine the scattering amplitudes at $t = 0$ and $t = 1$. For this, we solve the spatial equation involving the matrix $U$ (79). Let us first write this equation at $t = 0$. For the second component of $\phi = (\phi_1 \, \phi_2)^{\text{T}}$, we get

$$\partial_x[e^{-ikx}\phi_2(x,0)] = \delta(x)\phi_1(x,0), \tag{84}$$

and the same equation holds for $\bar{\phi}_2$. Integrating this equation with the boundary conditions at $-\infty$ (80), we obtain

$$e^{-ikx}\phi_2(x,0) = \Theta(x)\phi_1(0,0), \quad e^{-ikx}\bar{\phi}_2(x,0) = -1 + \Theta(x)\bar{\phi}_1(0,0). \tag{85}$$

Using these expressions in the equations for the first components $\phi_1$ and $\bar{\phi}_1$ (79) yields

$$\partial_x[e^{ikx}\phi_1(x,0)] = \bar{\Omega}(x)e^{2ikx}\Theta(x)\phi_1(0,0), \tag{86a}$$

$$\partial_x[e^{ikx}\bar{\phi}_1(x,0)] = \bar{\Omega}(x)e^{2ikx}[-1 + \Theta(x)\bar{\phi}_1(0,0)]. \tag{86b}$$

Integrating with the boundary conditions (80), we obtain

$$e^{ikx}\phi_1(x,0) = 1 + \Theta(x)\phi_1(0,0)\int_0^x \bar{\Omega}(x')e^{2ikx'}\mathrm{d}x', \tag{87}$$

$$e^{ikx}\bar{\phi}_1(x,0) = -\int_{-\infty}^x \bar{\Omega}(x')e^{2ikx'}\mathrm{d}x' + \Theta(x)\bar{\phi}_1(0,0)\int_0^x \bar{\Omega}(x')e^{2ikx'}\mathrm{d}x'. \tag{88}$$

From these expressions, we deduce the expressions at $x = 0$,

$$\phi_1(0,0) = 1, \quad \bar{\phi}_1(0,0) = -\int_{-\infty}^0 \bar{\Omega}(x')e^{2ikx'}dx'. \tag{89}$$

Combining the results (85,87,88) with the asymptotic behaviour (81), we deduce the scattering amplitudes

$$b(k,0) = 1, \tag{90a}$$

$$\bar{b}(k,0) = -\int_{-\infty}^{\infty} \bar{\Omega}(x)e^{2ikx}dx - \left(\int_{-\infty}^0 \bar{\Omega}(x)e^{2ikx}dx\right)\left(\int_0^{\infty} \bar{\Omega}(x')e^{2ikx'}dx'\right). \tag{90b}$$

The amplitudes $a(k,0)$ and $\bar{a}(k,0)$ can also be determined, but we will not need them in the following, so we do not write their expressions explicitly.

We can proceed similarly at final time $t = 1$, this time starting with the equations for the first components, as it is the one that involves the $\delta$ functions,

$$\partial_x[e^{ikx}\phi_1(x,1)] = (\alpha\delta(x) + \beta e^{2ik\xi}\delta(x-\xi))e^{-ikx}\phi_2(x,1), \tag{91}$$

with again the same equation for $\bar{\phi}_2$. Integrating with the asymptotic at $-\infty$ (80) yields

$$e^{ikx}\phi_1(x,1) = e^{2k^2} + \alpha\phi_2(0,1)\Theta(x) + \beta e^{ik\xi}\phi_2(\xi,1)\Theta(x-\xi), \tag{92a}$$

$$e^{ikx}\bar{\phi}_1(x,1) = \alpha\bar{\phi}_2(0,1)\Theta(x) + \beta e^{ik\xi}\bar{\phi}_2(\xi,1)\Theta(x-\xi). \tag{92b}$$

Inserting these expressions into the equations for $\phi_2$ and $\bar{\phi}_2$, we get

$$\partial_x[e^{-ikx}\phi_2(x,1)] = \Omega(x)e^{-2ikx}\left[e^{2k^2} + \alpha\phi_2(0,1)\Theta(x) + \beta e^{ik\xi}\phi_2(\xi,1)\Theta(x-\xi)\right], \tag{93a}$$

$$\partial_x[e^{-ikx}\bar{\phi}_2(x,1)] = \Omega(x)e^{-2ikx}\left[\alpha\bar{\phi}_2(0,1)\Theta(x) + \beta e^{ik\xi}\bar{\phi}_2(\xi,1)\Theta(x-\xi)\right]. \tag{93b}$$

Integrating with the asymptotic behaviour (80), we obtain

$$e^{-ikx}\phi_2(x,1) = e^{2k^2}\int_{-\infty}^x \Omega(x')e^{-2ikx'}dx' + \alpha\Theta(x)\phi_2(0,1)\int_0^x \Omega(x')e^{-2ikx'}dx'$$
$$+ \beta\Theta(x-\xi)\phi_2(\xi,1)e^{ik\xi}\int_{\xi}^x e^{-2ikx'}\Omega(x')dx', \tag{94}$$

$$e^{-ikx}\bar{\phi}_2(x,1) = -e^{-2k^2} + \alpha\Theta(x)\bar{\phi}_2(0,1)\int_0^x \Omega(x')e^{-2ikx'}dx'$$
$$+ \beta\Theta(x-\xi)\bar{\phi}_2(\xi,1)e^{ik\xi}\int_{\xi}^x e^{-2ikx'}\Omega(x')dx'. \tag{95}$$

We therefore deduce

$$\phi_2(0,1) = e^{2k^2}\int_{-\infty}^0 \Omega(x')e^{-2ikx'}dx', \tag{96}$$

$$\phi_2(\xi,1)e^{-ik\xi} = e^{2k^2}\int_{-\infty}^{\xi} \Omega(x')e^{-2ikx'}dx'$$
$$+ e^{2k^2}\alpha\left(\int_{-\infty}^0 \Omega(x')e^{-2ikx'}dx'\right)\left(\int_0^{\xi} \Omega(x')e^{-2ikx'}dx'\right), \tag{97}$$

$$\bar{\phi}_2(0,1) = -e^{-2k^2}, \quad \bar{\phi}_2(\xi,1)e^{-ik\xi} = -e^{-2k^2}\left(1 + \alpha\int_0^{\xi} \Omega(x')e^{-2ikx'}dx'\right). \tag{98}$$

From the solutions (92a,94), we can read the asymptotic behaviours (81) and deduce the scattering amplitudes

$$b(k,1)e^{-2k^2} = \int_{-\infty}^{+\infty} \Omega(x')e^{-2ikx'}dx' + \alpha\left(\int_{-\infty}^{0} \Omega(x')e^{-2ikx'}dx'\right)\left(\int_{0}^{+\infty} \Omega(x')e^{-2ikx'}dx'\right)$$

$$+ \beta e^{2ik\xi}\left(\int_{-\infty}^{\xi} \Omega(x')e^{-2ikx'}dx'\right)\left(\int_{\xi}^{+\infty} \Omega(x')e^{-2ikx'}dx'\right)$$

$$+ \alpha\beta e^{2ik\xi}\left(\int_{-\infty}^{0} \Omega(x')e^{-2ikx'}dx'\right)\left(\int_{0}^{\xi} \Omega(x')e^{-2ikx'}dx'\right)\left(\int_{\xi}^{+\infty} \Omega(x')e^{-2ikx'}dx'\right), \quad (99)$$

$$\bar{b}(k,1)e^{2k^2} = -\alpha - \beta e^{2ik\xi}\left(1 + \alpha\int_{0}^{\xi} \Omega(x')e^{-2ikx'}dx'\right). \quad (100)$$

Again, $a$ and $\bar{a}$ can be obtained similarly but we will not need them here.

The last step is to relate the scattering amplitudes $b$ and $\bar{b}$ at initial time (90a) with the ones at final time $t = 1$ (99,100) by using the time evolution (83a). This gives the following equations for $\Omega$ and $\bar{\Omega}$:

$$\int_{-\infty}^{+\infty} \Omega(x')e^{-2ikx'}dx' + \alpha\left(\int_{-\infty}^{0} \Omega(x')e^{-2ikx'}dx'\right)\left(\int_{0}^{+\infty} \Omega(x')e^{-2ikx'}dx'\right)$$

$$+ \alpha\beta e^{2ik\xi}\left(\int_{-\infty}^{0} \Omega(x')e^{-2ikx'}dx'\right)\left(\int_{0}^{\xi} \Omega(x')e^{-2ikx'}dx'\right)\left(\int_{\xi}^{+\infty} \Omega(x')e^{-2ikx'}dx'\right)$$

$$+ \beta e^{2ik\xi}\left(\int_{-\infty}^{\xi} \Omega(x')e^{-2ikx'}dx'\right)\left(\int_{\xi}^{+\infty} \Omega(x')e^{-2ikx'}dx'\right) = e^{-4k^2}, \quad (101)$$

$$\int_{-\infty}^{\infty} \bar{\Omega}(x)e^{2ikx}dx + \left(\int_{-\infty}^{0} \bar{\Omega}(x)e^{2ikx}dx\right)\left(\int_{0}^{\infty} \bar{\Omega}(x')e^{2ikx'}dx'\right)$$

$$= e^{-4k^2}\left(\alpha + \beta e^{2ik\xi} + \alpha\beta e^{2ik\xi}\int_{0}^{\xi} \Omega(x')e^{-2ikx'}dx'\right). \quad (102)$$

We can obtain equations in real space by taking the inverse Fourier transform. More precisely, multiplying (101) by $e^{2ikx}/\pi$ and integrating over $k$ yields the equation for $\Omega$ (8). Similarly, multiplying (102) by $e^{-2ikx}/\pi$ and integrating over $k$, we obtain the equation for $\bar{\Omega}$ (9).

These integral equations (8,9) clearly admit a unique solution when $\alpha = \beta = 0$, given by $\Omega(x) = K(x)$ and $\bar{\Omega}(x) = 0$. One can then use this starting point to look for a perturbative solution for small $\alpha$ and $\beta$, as it is done in Section 6 below. This leads us to expect that these equations admit a unique solution, at least for small $\alpha$ and $\beta$.

This concludes our derivation of the integral equations (8,9). There only remains to determine the constants $\alpha$ and $\beta$.

## 5.3 Determination of the remaining constants $\alpha$ and $\beta$

We now turn to the determination of the last constants $\alpha$ and $\beta$ which appear in the integral equations (8,9). A first equation can be obtained from the conservation of the number of particles in the SEP between initial and final time, i.e.,

$$\int_{-\infty}^{\infty} [\Phi(x) - \bar{\Phi}(x)]dx = 0. \quad (103)$$

This equation can be derived from the MFT equation (42), by integration on $x$ from $-\infty$ to $+\infty$, which yields $\int_{-\infty}^{\infty} \partial_t q(x,t)\mathrm{d}x = 0$, since $\partial_x p$ and $\partial_x q$ decay to 0 at $\pm\infty$.

The second equation can be determined by following the approach used in [8], which relies on the scattering formalism. The scattering amplitudes defined in (81) can be equivalently defined by regrouping the two vectors $\phi$ and $\bar{\phi}$ in a single matrix, so that

$$\begin{pmatrix} a(k,t) & \bar{b}(k,t) \\ b(k,t) & -\bar{a}(k,t) \end{pmatrix} = \lim_{x\to\infty}\lim_{y\to-\infty} M(x,t;k)M(y,t;k)^{-1} \begin{pmatrix} e^{2k^2 t} & 0 \\ 0 & -e^{-2k^2 t} \end{pmatrix}, \qquad (104)$$

where the matrix $M(x;k,t)$ satisfies

$$\partial_x M(x,t;k) = U M(x,t;k), \quad \partial_t M(x,t;k) = V M(x,t;k), \qquad (105)$$

with the matrices $U$ and $V$ given in (79). Remarkably, the spatial equation for $M$ can be explicitly solved when $k = 0$ by using the specific form of the functions $u$ and $v$ in terms of $p$ and $q$ (68). The solution is given in the Supplemental Material of Ref. [8], and reads

$$M(x,t;k) = \begin{pmatrix} \sqrt{K} & 0 \\ 0 & \frac{1}{\sqrt{K}} \end{pmatrix} \begin{pmatrix} e^{\int_{-\infty}^{x}(1-q)\partial_x p} & e^{-\int_{-\infty}^{x} q\partial_x p} \\ -(1-\rho)e^{\int_{-\infty}^{x} q\partial_x p} & \rho\, e^{-\int_{-\infty}^{x}(1-q)\partial_x p} \end{pmatrix} \begin{pmatrix} \frac{1}{\sqrt{K}} & 0 \\ 0 & \sqrt{K} \end{pmatrix}, \qquad (106)$$

where $K$ is the constant we introduced above Eq. (73) to get rid of the constant in front of the $\delta$ function in the initial condition for $u$. Using the asymptotic behaviours $q(x,t) \xrightarrow{t\to\pm\infty} \rho_\pm$, $p(x,t) \xrightarrow{t\to-\infty} 0$ and $p(x,t) \xrightarrow{t\to+\infty} \lambda + \nu$, we get

$$\lim_{t\to+\infty} M(t) = \begin{pmatrix} C\, e^{\lambda+\mu} & C \\ -(1-\rho_+)C^{-1} & \rho_+\, C^{-1} e^{-\lambda-\mu} \end{pmatrix}, \quad \lim_{t\to-\infty} M(t) = \begin{pmatrix} 1 & 1 \\ -(1-\rho_-) & \rho_+ \end{pmatrix}, \quad (107)$$

where we have denoted $C = e^{-\int_{-\infty}^{\infty} q\partial_x p}$. Using these expressions in (104), we obtain a simple expression for the product of the diagonal elements at $k = 0$,

$$-b(0,t)\bar{b}(0,t) = \omega \equiv \rho_+(e^{-\lambda-\nu}-1) + \rho_-(e^{\lambda+\nu}-1) + \rho_+\rho_-(e^{\lambda+\nu}-1)(e^{-\lambda-\nu}-1), \quad (108)$$

with $\omega$ the single parameter identified for the SEP in [9,33], still with two parameters $\lambda$ and $\nu$. Using the expressions of $b$ and $\bar{b}$ at $t = 0$ (90a), and the bulk equation for $\bar{\Omega}$ (102), this last equation yields

$$\alpha + \beta + \alpha\beta \int_0^\xi \Omega(x')\mathrm{d}x' = \omega. \qquad (109)$$

Note that the same equation can be obtained from the expressions at $t = 1$ (99,100). Comparing with the definition of the kernel $\bar{K}$ (10), we notice that this equation can also be written in the compact form

$$\int_{-\infty}^{+\infty} \bar{K}(x)\mathrm{d}x = \omega. \qquad (110)$$

With this last derivation, we now have all the equations needed to determine the profiles $\Phi$ and $\bar{\Phi}$ and thus deduce the cumulants from (45).

## 6 Perturbative solution for the first joint cumulants

We do not have an explicit solution of the equation for $\Omega$ (8), so we will rely on a perturbative solution in $\lambda$ and $\nu$.

## 6.1 For the currents

The equations for $\Omega$ (8) and $\bar{\Omega}$ (9) only involve the parameters $\alpha$ and $\beta$. We thus first write the solutions of these equations perturbatively in $\alpha$ and $\beta$, and in a second step express them in terms of $\lambda$ and $\nu$ by using relations (11)-(16). We denote the expansions of $\Omega$ and $\bar{\Omega}$ as

$$\Omega(x) = \sum_{n,m=0}^{\infty} \alpha^n \beta^m \, \Omega_{n,m}(x), \quad \bar{\Omega}(x) = \sum_{n,m=0}^{\infty} \alpha^n \beta^m \, \bar{\Omega}_{n,m}(x). \tag{111}$$

Inserting these expansions into the integral equations (8,9), we obtain

$$\Omega_{0,0}(x) = K(x) = \frac{e^{-\frac{x^2}{4}}}{\sqrt{4\pi}}, \tag{112}$$

$$\Omega_{1,0}(x) = -\frac{e^{-\frac{x^2}{8}}}{4\sqrt{2\pi}} \operatorname{erfc}\left(\frac{|x|}{2\sqrt{2}}\right), \quad \Omega_{0,1}(x) = -\frac{e^{-\frac{(x+\xi)^2}{4}}}{4\sqrt{2\pi}} \operatorname{erfc}\left(\frac{|x-\xi|}{2\sqrt{2}}\right), \tag{113}$$

$$\bar{\Omega}_{0,0}(x) = 0, \quad \bar{\Omega}_{1,0}(x) = \frac{e^{-\frac{x^2}{4}}}{\sqrt{4\pi}}, \quad \bar{\Omega}_{0,1}(x) = \frac{e^{-\frac{(x-\xi)^2}{4}}}{\sqrt{4\pi}}, \tag{114}$$

$$\bar{\Omega}_{2,0}(x) = -\frac{e^{-\frac{x^2}{8}}}{4\sqrt{2\pi}} \operatorname{erfc}\left(\frac{|x|}{2\sqrt{2}}\right), \quad \bar{\Omega}_{0,2}(x) = -\frac{e^{-\frac{(x-2\xi)^2}{8}}}{4\sqrt{\pi}} \operatorname{erfc}\left(\frac{|x|}{2\sqrt{2}}\right), \tag{115}$$

$$\bar{\Omega}_{1,1}(x) = -\frac{e^{-\frac{(x-\xi)^2}{8}}}{2\sqrt{\pi}} \operatorname{erfc}\left(\frac{|x|+\xi}{2\sqrt{2}}\right). \tag{116}$$

To deduce $\Phi$ and $\bar{\Phi}$, we integrate $\Omega$ and $\bar{\Omega}$ (7) with respect to $x$, with the boundary conditions at infinity (15),

$$\Phi(x) = \begin{cases} \rho_- + \frac{1}{a_-} \int_{-\infty}^x \Omega, & \text{for } x < 0, \\ d_0 + \frac{1}{a_0} \int_0^x \Omega, & \text{for } 0 < x < \xi, \\ \rho_+ - \frac{1}{a_+} \int_x^\infty \Omega, & \text{for } x > \xi, \end{cases} \quad \bar{\Phi}(x) = \begin{cases} \rho_- + \frac{1}{b_-} \int_{-\infty}^x \bar{\Omega}, & \text{for } x < 0, \\ \rho_+ - \frac{1}{b_+} \int_x^\infty \bar{\Omega}, & \text{for } x > 0. \end{cases} \tag{117}$$

For the above expressions, these integrals can be computed using the tables in [43]. We will also need the integral of $\Phi$ and $\bar{\Phi}$ in Eqs. (16,17), which correspond to double integrals of $\Omega$. This is not convenient to compute in practice, and it is more practical to use integration by parts

$$\int_x^\infty \mathrm{d}y \int_y^\infty \mathrm{d}z \, \Omega(z) = -x \int_x^\infty \Omega(y)\mathrm{d}y + \int_x^\infty y\Omega(y)\mathrm{d}y, \tag{118}$$

which can now be computed using the tables in [43].

Next, we expand all the parameters in powers of $\lambda$ and $\nu$,

$$Z = \sum_{n,m \geq 0} Z_{n,m} \lambda^n \nu^m, \tag{119}$$

with $Z \in \{\alpha, \beta, a_-, a_0, a_+, b_-, b_+, d_0\}$. Inserting these expansions into the boundary conditions at $x = 0$ and $x = \xi$ (11,13) and into the conservation equations (17,16), we obtain the coefficients of these expansions up to order 4 included for $\alpha$ and $\beta$ and up to order 3 included for $b_+$, $b_-$, $d_0$, $\frac{1}{a_-}$, $\frac{1}{a_0}$ and $\frac{1}{a_+}$ in the case $\rho_+ = \rho_- = \rho$. This difference of orders come from the fact that $\alpha$ and $\beta$ begin at order 2 in $\lambda$ and $\nu$,

$$\alpha = \lambda(\lambda + \nu)\rho(1-\rho) + \cdots, \quad \beta = \nu(\lambda + \nu)\rho(1-\rho) + \cdots, \tag{120}$$

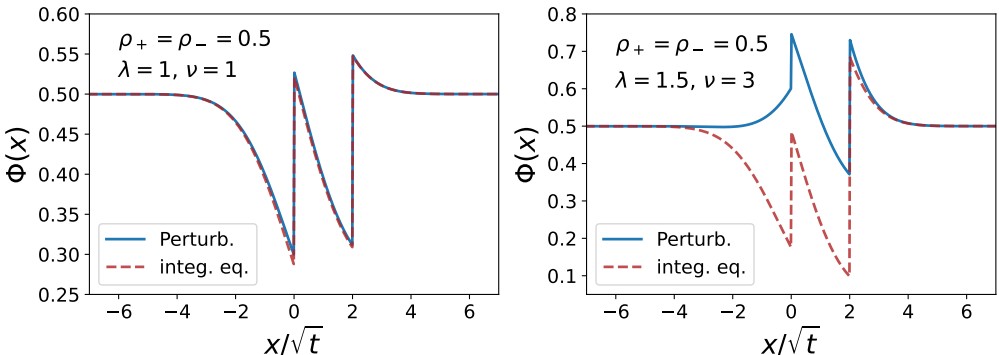

Figure 3: Profile $\Phi$ at final time obtained from the numerical solution of the integral equations (8,9) with the conditions (11-17) (dashed red lines), compared to the perturbative solution (121) up to order 4 in $\lambda$ and $\nu$ (solid blue line). Remarkably, the perturbative solution is in good agreement even for values $\lambda = \nu = 1$ which are not small (left), but ultimately differs from the correct solution when $\lambda$ or $\nu$ is increased (right).

therefore so does $\bar{\Omega}$, while $\Omega$ already has non zero terms at order 0. On the other hand, $\Phi$ and $\bar{\Phi}$ have terms at first order in $\lambda$ and $\nu$. This is why the expansions of $b_+$ and $b_-$ begin at order 1, while those of $a_+$, $a_0$ and $a_-$ have terms in $1/\lambda$ or $1/\nu$.

Consequently, we obtain the lowest orders of the profiles $\Phi$ and $\bar{\Phi}$. For instance,

$$
\begin{aligned}
\Phi(x > \xi) = {}& \rho + \frac{1}{2}(\lambda + \nu)\rho(1-\rho)\,\mathrm{erfc}\left(\frac{x}{2}\right) + \frac{1}{4}(\lambda + \nu)^2\rho(1-\rho)(1-2\rho)\,\mathrm{erfc}\left(\frac{x}{2}\right) \\
& + \frac{1}{24}\rho(1-\rho)(\lambda + \nu)^2\left[2\,\mathrm{erfc}\left(\frac{x}{\sqrt{2}}\right)\left((\lambda + \nu)(1-3\rho(1-\rho)) + 3\nu\rho(1-\rho)\,\mathrm{erf}\left(\frac{\xi}{2}\right)\right)\right. \\
& \left. -3\rho(1-\rho)\left(8\nu\mathrm{T}\left(\frac{\xi}{\sqrt{2}},\frac{x}{\xi}\right) + 8\nu\mathrm{T}\left(\frac{\xi+x}{2}, 1 - \frac{2x}{\xi+x}\right) + 2\nu\,\mathrm{erf}\left(\frac{\xi}{2}\right)\right.\right. \\
& \left.\left. -2\nu\,\mathrm{erf}\left(\frac{\xi+x}{2\sqrt{2}}\right) + \lambda\,\mathrm{erfc}\left(\frac{x}{2\sqrt{2}}\right)^2\right)\right] + \mathcal{O}(\lambda^4, \nu^4), \quad (121)
\end{aligned}
$$

where $\mathrm{T}(z, a)$ is Owen's T-function, defined by [43]

$$
T(z, a) = \frac{1}{2\sqrt{2\pi}}\int_{-z}^{\infty} e^{-\frac{t^2}{2}}\,\mathrm{erf}\left(\frac{at}{\sqrt{2}}\right)\mathrm{d}t\,. \tag{122}
$$

We compare this perturbative expression to the result obtained from the numerical resolution of the integral equations (8,9) in Fig. 3. Although the perturbative solution is expected to be in good agreement with the actual solution for values of $\lambda$ and $\nu$ which are small, we observe a reasonnable agreement up to $\lambda = \nu = 1$. To accurately describe larger values of $\lambda$ or $\nu$, one should include higher orders the perturbation series.

From the expressions of $\Phi$ and $\bar{\Phi}$, we deduce $\hat{\psi}_\xi$ from (19), which yields

$$
\hat{\psi}_\xi(\lambda, v) = -v\xi\rho + \rho(1-\rho)\left[\frac{1}{2}v(\lambda+v)\left(\frac{2e^{-\frac{\xi^2}{4}}}{\sqrt{\pi}} - \xi\,\mathrm{erfc}\left(\frac{\xi}{2}\right)\right) + \frac{\lambda(\lambda+v)}{\sqrt{\pi}} + \frac{v^2\xi}{2}\right]
$$

$$
+ v\rho(1-\rho)(1-2\rho)\left[\frac{\lambda(\lambda+v)}{4}\left(\frac{2e^{-\frac{\xi^2}{4}}}{\sqrt{\pi}} - \xi\,\mathrm{erfc}\left(\frac{\xi}{2}\right)\right) - \frac{\lambda(\lambda+v)}{2\sqrt{\pi}} - \frac{v^2\xi}{6}\right]
$$

$$
+ \frac{\rho(1-\rho)}{24}\left[v(\lambda+v)(2\lambda^2+\lambda v+v^2)\left(\frac{2e^{-\frac{\xi^2}{4}}}{\sqrt{\pi}} - \xi\,\mathrm{erfc}\left(\frac{\xi}{2}\right)\right)\right.
$$

$$
\left. + \frac{2}{\sqrt{\pi}}\lambda(\lambda+v)(\lambda^2+\lambda v+2v^2+\mu^4\xi)\right]
$$

$$
+ \frac{\rho^2(1-\rho^2)}{4}\left[\lambda v(v^2-\lambda^2)\left(\frac{2e^{-\frac{\xi^2}{4}}}{\sqrt{\pi}} - \xi\,\mathrm{erfc}\left(\frac{\xi}{2}\right)\right) - \frac{\lambda(\lambda+v)(\sqrt{2}\lambda^2+\sqrt{2}\lambda v+4v^2)}{\sqrt{\pi}}\right.
$$

$$
- \frac{\lambda v(\lambda+v)^2}{2}\left(\frac{8e^{-\frac{\xi^2}{8}}}{\sqrt{2\pi}} - \xi\,\mathrm{erfc}\left(\frac{\xi}{2\sqrt{2}}\right)\right)\mathrm{erfc}\left(\frac{\xi}{2\sqrt{2}}\right) - \xi v^4
$$

$$
\left. - v^2(\lambda+v)^2\left(\frac{2e^{-\frac{\xi^2}{2}}}{\sqrt{2\pi}} - \xi\,\mathrm{erfc}\left(\frac{\xi}{\sqrt{2}}\right)\right) + \frac{2\lambda v(\lambda+v)^2}{\sqrt{\pi}}\mathrm{erfc}\left(\frac{\xi}{2}\right)\right] + \mathcal{O}(\lambda^5, v^5). \quad (123)
$$

One can check that, for $v = 0$, we recover the first orders[5] of the cumulant generating function of [9], for $\lambda = 0$ it gives the first orders of the one obtained in [3], and for $\xi = 0$, since $J_t = Q_t$, we recover the first orders of the one for $Q_t$ [9], evaluated at $\lambda + v$. Taking derivatives of this expression with respect to $\lambda$ and $v$, we obtain the different joint cumulants $\langle Q_t^n J_t^m \rangle_c$ of the two currents,

$$
\hat{\psi}_\xi(\lambda, v) = \sum_{n,m\geq 0} \frac{\lambda^n}{n!}\frac{v^m}{m!}\langle Q_t^n J_t^m \rangle_c. \quad (124)
$$

The first cumulants are written in Section 2.

## 6.2 For the current/tracer correlations

The distribution of the position of a tracer can be obtained from the distribution of the current $J_t$ [3, 4]. The idea is that, since the particles remain in the same order, the number of particles to the right of the tracer is conserved, the tracer is located at the position $X_t$ such that $J_t(X_t) = 0$. This relation is not quite exact, since there could be several values for which $J_t(x) = 0$. Actually, $X_t$ corresponds to the smallest of these values. However, in the long time limit, this indeterminacy becomes a subdominant correction, and this relation becomes exact at leading order in $t$. This implies that $\mathbb{P}(X_t = x) = \mathbb{P}(J_t(x) = 0)$. We can directly extend this relation to the joint distribution of $Q_t$ and $X_t$,

$$
\mathbb{P}(Q_t = q\ \&\ X_t = x) = \mathbb{P}(Q_t = q\ \&\ J_t(x) = 0). \quad (125)
$$

---

[5]In fact, not only the first orders, but the full cumulant generating function of [9] is recovered from (8) which can be solved in this case. Similar comments apply to the specific cases $\lambda = 0$ and $\xi = 0$.

We have computed the joint cumulant generating function of the two currents, which implies

$$\left\langle e^{\lambda Q_t + \nu J_t(x_t)} \right\rangle = \sum_q \sum_j e^{\lambda q + \nu j} \, \mathbb{P}(Q_t = q \,\&\, J_t(x) = j) \underset{t \to \infty}{\simeq} e^{\sqrt{t}\hat{\psi}_\xi(\lambda, \nu)}. \tag{126}$$

We can take the inverse Laplace transform in $\nu$, which can be evaluated by a saddle point approximation for large $t$, which gives a mixed distribution/generating function

$$\left\langle e^{\lambda Q_t} \delta(J_t(x_t) - j\sqrt{t}) \right\rangle \underset{t \to \infty}{\simeq} e^{-\sqrt{t}\varphi_\xi(\lambda, j)}, \tag{127}$$

where $\varphi_\xi$ is given by the Legendre transform

$$\varphi_\xi(\lambda, j) = \nu^\star(\lambda, j)j - \hat{\psi}_\xi(\lambda, \nu^\star(\lambda, j)), \quad \partial_\nu \hat{\psi}_\xi(\lambda, \nu)\big|_{\nu^\star} = j. \tag{128}$$

Using the relation between $J_t$ and $X_t$ (125), we deduce

$$\left\langle e^{\lambda Q_t} \delta(X_t - \xi\sqrt{t}) \right\rangle \underset{t \to \infty}{\simeq} e^{-\sqrt{t}\varphi_\xi(\lambda, 0)}. \tag{129}$$

We can obtain the joint cumulant generating function of the current $Q_t$ and the position $X_t$ by another Legendre transform,

$$\lim_{t \to \infty} \frac{1}{\sqrt{t}} \ln \left\langle e^{\lambda Q_t + \chi X_t} \right\rangle = \chi \xi^\star(\lambda, \chi) - \varphi_{\xi^\star(\lambda, \chi)}(\lambda, 0), \quad \partial_\xi \varphi_\xi(\lambda, 0)\big|_{\xi^\star} = \chi. \tag{130}$$

This procedure extends the one of [3, 4] to the joint distribution of $Q_t$ and $X_t$. It can be carried out explicitly starting from the expression of $\hat{\psi}_\xi$ at lowest orders (123), and yields

$$\begin{aligned}
\lim_{t \to \infty} \frac{1}{\sqrt{t}} \ln \left\langle e^{\lambda Q_t + \chi X_t} \right\rangle &= \frac{1-\rho}{\rho\sqrt{\pi}}(\chi + \lambda\rho)^2 - \frac{(1-\rho)^2}{\rho\sqrt{\pi}}\lambda\chi(\chi + \lambda\rho) + \frac{\lambda^3(1-\rho)(\lambda\rho + \chi)}{12\sqrt{\pi}} \\
&\quad - \frac{\lambda^3\chi(1-\rho)^3}{4\sqrt{\pi}} - \frac{\lambda^2\chi(\lambda\rho + \chi)(1-\rho)^3}{4\sqrt{\pi}\rho^2} + \frac{\lambda^2\chi(\lambda\rho + \chi)(1-\rho)^2}{4\sqrt{\pi}} \\
&\quad - \frac{\chi^3(\lambda\rho + \chi)(1-\rho)}{4\sqrt{\pi}\rho^3} + \frac{\chi(\lambda\rho + \chi)^3(1-\rho)^3}{\pi^{3/2}\rho^3} \\
&\quad + \frac{2\chi^3(\lambda\rho + \chi)(1-\rho)^2}{3\sqrt{\pi}\rho^2} + \frac{\chi^3(\lambda\rho + \chi)(1-\rho)}{3\sqrt{\pi}\rho^2} - \frac{(\lambda\rho + \chi)^4(1-\rho)^2}{2\sqrt{2\pi}\rho^2} \\
&\quad - \frac{\lambda\chi^2(\lambda\rho + \chi)(1-\rho)^2(1-2\rho)}{2\sqrt{\pi}\rho^2} + \frac{\lambda\chi(\lambda + \chi)(\lambda\rho + \chi)(1-\rho)}{4\sqrt{\pi}\rho^2} \\
&\quad + \mathcal{O}(\lambda^5, \chi^5).
\end{aligned} \tag{131}$$

This directly gives the first joint cumulants of $Q_t$ and $X_t$, which are given in Section 2. In particular, setting $\chi = -\rho\lambda$, this gives the generating function of $Q_t - \rho X_t$,

$$\lim_{t \to \infty} \frac{1}{\sqrt{t}} \ln \left\langle e^{\lambda(Q_t - \rho X_t)} \right\rangle = \frac{\rho(1-\rho)^3}{4\sqrt{\pi}}\lambda^4 + \mathcal{O}(\lambda^5). \tag{132}$$

Remarkably, there is no term in $\lambda^2$, indicating that the variance of $Q_t - \rho X_t$ is smaller than $\sqrt{t}$ for large $t$, indicating strong correlations between the current $Q_t$ and the positions $X_t$ of the tracer. However, this does not indicate that $Q_t$ and $X_t$ are proportional, since the higher order cumulants grow as $\sqrt{t}$. This indicates that $\text{Var}(Q_t - \rho X_t)$ is nonzero, but grows slower than $\sqrt{t}$.

## 7 Conclusion

We have studied the joint distribution of the current $Q_t$ through the origin and the current $J_t$ through a moving boundary in the SEP, as well as their correlations with the density of particles. These correlations are described by generalised density profiles. We have obtained integral equations satisfied by these generalised density profiles. These integral equations extend the ones discovered in [6,7] in the case of a single observable (current $Q_t$ or $J_t$, tracer position $X_t$). This further emphasises the key role of such strikingly simple integral equations involving partial convolutions in interacting particle systems. In the case of a single observable, the integral equations naturally obtained are bilinear, but surprisingly they are equivalent to *linear* equations at the expense of introducing analytic continuations [6,7]. An important open question is whether the equation obtained here, which is trilinear, can be reduced to such linear equations (for which an explicit solution can be obtained).

We have also obtained simple boundary conditions for the generalised density profiles. These boundary conditions take a simple physical form, in terms of the chemical potential, and can be applied to any model of single-file diffusion. This extends the relations that have been obtained for the SEP from microscopic considerations [5–7].

As a consequence of these equations, we have characterised the joint statistics of the current through the origin $Q_t$ and the position of a tracer $X_t$, initially at the origin. These variables are strongly correlated, and even become equal in the high density limit.

This work opens the way to the study of more than two observables, such as multiple currents or tracers, in the SEP and other models of single-file systems.

## Acknowledgments

We thank Alexis Poncet for illuminating discussions and in particular for sharing his guess of the Eq. (33), left, involving the pressure.

## A  Mapping the boundary conditions for other observables

In this Appendix, we show that the boundary conditions (11,12) obtained for the currents $Q_t$ and $J_t$ can be mapped onto other physical boundary conditions for other observables, such as the position $X_t$ of a tracer. In Ref. [34], it has been shown that the current $Q_t$ in a single-file system described by the coefficients $D(\rho)$ ans $\sigma(\rho)$ corresponds to the opposite of the displacement of a tracer in a dual single-file system, with

$$\tilde{D}(\rho) = \frac{1}{\rho^2} D\left(\frac{1}{\rho}\right), \quad \tilde{\sigma}(\rho) = \rho\, \sigma\left(\frac{1}{\rho}\right). \tag{A.1}$$

The mapping is as follows. The density $\tilde{\rho}$ in the dual system, in the reference frame of the tracer at $\tilde{X}_t = -Q_t$, can be expressed in terms of the density $\rho$ of the initial system as [34]

$$\rho(x,t) = \frac{1}{\tilde{\rho}(k(x,t),t)}, \quad k(x,t) = \int_0^x \rho(y,t) \mathrm{d}y. \tag{A.2}$$

Since this mapping is valid for all realisations of $\rho$, it is also valid for the saddle point $(q,p)$ solution of the MFT equations (42,43), and thus for the profile $\Phi$ (47),

$$\Phi(x) = \frac{1}{\tilde{\Phi}(z(x))}, \quad z(x) = \int_0^x \Phi(y) \mathrm{d}y. \tag{A.3}$$

The dual profile $\tilde{\Phi}$ corresponds to [34]

$$\frac{\left\langle \tilde{\rho}(\tilde{X}_t + r, t) e^{-\lambda \tilde{X}_t} \right\rangle}{\left\langle e^{-\lambda \tilde{X}_t} \right\rangle} \underset{t \to \infty}{\simeq} \tilde{\Phi}\left( z = \frac{r}{\sqrt{t}} \right), \tag{A.4}$$

with an unusual minus sign in the exponential, due to the fact that $\tilde{X}_t = -Q_t$.

The chemical potential (32) becomes

$$\mu(\rho) = \int^\rho \frac{2D(r)}{\sigma(r)} dr = -\int^{\frac{1}{\rho}} \frac{2D(1/r)}{\sigma(1/r)} \frac{dr}{r^2} = -\int^{\frac{1}{\rho}} \frac{2r\tilde{D}(r)}{\tilde{\sigma}(r)} dr = -\tilde{P}\left( \frac{1}{\rho} \right), \tag{A.5}$$

where $\tilde{P}$ is the pressure (34) for the dual system.

Combining these relations with (12), we obtain

$$\tilde{P}(\tilde{\Phi}(0^+)) - \tilde{P}(\tilde{\Phi}(0^-)) = -\lambda. \tag{A.6}$$

Changing $\lambda \to -\lambda$ to remove the minus sign in the definition (A.4), we obtain (33), left. For the relation involving the derivative, we need

$$\partial_x \mu(\Phi) = \frac{2D(\phi)\partial_x \Phi}{\sigma(\Phi)} = \frac{2\tilde{\Phi}^3 \tilde{D}(\tilde{\Phi})}{\tilde{\sigma}(\tilde{\Phi})} \frac{\partial z}{\partial x} \partial_z \left( \frac{1}{\tilde{\Phi}} \right) = -\frac{2\tilde{D}(\tilde{\Phi})}{\tilde{\sigma}(\tilde{\Phi})} \partial_z \tilde{\Phi} = -\partial_z \tilde{\mu}(\tilde{\Phi}). \tag{A.7}$$

From (12), we thus straightforwardly deduce (33), right.

## B Relation between the physical boundary conditions and the microscopic equations for the SEP

In the case of the SEP, other boundary conditions have been obtained for the different observables considered here (individually), from microscopic considerations [5–7]. Note that they have been obtained with a different choice for the time scale, with $D(\rho) = \frac{1}{2}$ and $\sigma(\rho) = \rho(1-\rho)$. Here, we show that they are equivalent to the physical boundary conditions (11,12,33) obtained in this article.

For the case of the current $Q_t$, they take the form [6]

$$\frac{\Phi(0^+)(1 - \Phi(0^-))}{\Phi(0^-)(1 - \Phi(0^+))} = e^\lambda, \quad \Phi'(0^\pm) = \mp 2\hat{\psi}\left( \frac{1}{1 - e^{\mp\lambda}} - \Phi(0^\pm) \right), \tag{B.1}$$

where $\Phi$ has been defined in [6] with a slightly different scaling

$$\frac{\left\langle \eta_r e^{\lambda Q_t} \right\rangle}{\left\langle e^{\lambda Q_t} \right\rangle} \underset{t \to \infty}{\simeq} \Phi\left( x = \frac{r}{\sqrt{2t}} \right). \tag{B.2}$$

Taking the logarithm of the first equation in (B.1), we get

$$\lambda = -\ln\left( \frac{1}{\Phi(0^+)} - 1 \right) + \ln\left( \frac{1}{\Phi(0^-)} - 1 \right), \tag{B.3}$$

which is exactly the relation (12) with the expression of the chemical potential for the SEP (14). Combining the relations (B.1) right to eliminate $\hat{\psi}$, and combining with the first relation to remove the $e^{\pm\lambda}$, we get

$$-\frac{\Phi'(0^+)(\Phi(0^+) - \Phi(0^-))}{\Phi(0^+)(1 - \Phi(0^+))} = \frac{\Phi'(0^-)(\Phi(0^-) - \Phi(0^+))}{\Phi(0^-)(1 - \Phi(0^-))}. \tag{B.4}$$

Rewriting it in terms of the chemical potential for the SEP (14), we obtain,

$$\mu'(\Phi(0^+))\Phi'(0^+) = \mu'(\Phi(0^-))\Phi'(0^-),\tag{B.5}$$

which is indeed Eq. (12).

In the case of a tracer at position $X_t$, different relations have been obtained [5] which read

$$\frac{1-\Phi(0^-)}{1-\Phi(0^+)} = e^\lambda, \quad \Phi'(0^\pm) = \mp\frac{2\hat{\psi}}{e^{\pm\lambda}-1}\Phi(0^\pm),\tag{B.6}$$

with the profiles defined as

$$\frac{\left\langle \eta_{X_t+r}e^{\lambda X_t}\right\rangle}{\left\langle e^{\lambda X_t}\right\rangle} \underset{t\to\infty}{\simeq} \Phi\left(x = \frac{r}{\sqrt{2t}}\right).\tag{B.7}$$

As for the current, taking the logarithm of the first equation in (B.6) yields

$$\lambda = -\ln(1-\Phi(0^+)) + \ln(1-\Phi(0^-)) = P(\Phi(0^+)) - P(\Phi(0^-)), \quad \text{with} \quad P(\rho) = -\ln(1-\rho),\tag{B.8}$$

for the SEP. This is indeed (33), left. Combining the relations in (B.6) to eliminate $\hat{\psi}$ and $\lambda$, we obtain

$$-\frac{\Phi'(0^+)}{\Phi(0^+)(1-\Phi(0^+))}(\Phi(0^+)-\Phi(0^-)) = \frac{\Phi'(0^-)}{\Phi(0^-)(1-\Phi(0^-))}(\Phi(0^-)-\Phi(0^+)).\tag{B.9}$$

This is identical to the case of the current (B.4), hence it yields again (33), right.

## C  Numerical resolution of the integral equation

To solve the integral equations (8,9), we discretize the integrals by using the trapezoidal rule

$$\int_a^b f(x)\mathrm{d}x = \frac{f(a)+f(b)}{2}\delta x + \sum_{k=1}^{N-1} f(a+k\,\delta x)\delta x,\tag{C.1}$$

where $N = \lfloor(b-a)/\delta x\rfloor$ and $\delta x$ is the discretization length. Furthermore, since $\lim_{x\to\pm\infty}\Omega(x) = 0$ and $\lim_{x\to\pm\infty}\bar{\Omega}(x) = 0$, we solve (8,9) on a finite interval $[-L,L]$ with the condition that $\Omega(x) = \bar{\Omega}(x) = 0$ if $x \notin [-L,L]$. By doing so, the integral equations (8,9) become a finite system of nonlinear equations in $\Omega(-L+i\,\delta x)$ and $\bar{\Omega}(-L+i\,\delta x)$ with $0 \le i \le N = \lfloor 2L/\delta x\rfloor$. This system can be solved using standard gradient descent algorithms.

The profiles $\Phi$ and $\bar{\Phi}$ are then obtained from these solutions by discrete integration of (7). The parameters $a_-$, $a_0$, $a_+$ and $b_-$, $b_+$ are determined from the boundary conditions (11,12,13). This procedure gives the profiles $\Phi$ and $\bar{\Phi}$, given $\alpha$ and $\beta$ as input. Performing a final gradient descent, $\alpha$ and $\beta$ are determined from the parameters $\lambda$, $\rho_-$ and $\rho_+$ from (16,17).

## D  Numerical resolution of the MFT equation

The coupled MFT equations have a forward/backward structure: $p$ obeys an antidiffusion equation (43) with a final condition (44) (left); while $q$ obeys a diffusion equation (42) with an initial condition (44) (right). To solve this system, we use the standard scheme described for instance in [11]. which we briefly summarize here.

1. First solve the equation for $q$ (42) using the initial guess $p(x,t) = \lambda\Theta(x) + \nu\Theta(x - \xi)$, corresponding to the terminal condition (44) (left) extended to all times.

2. Then, use the resulting solution $q(x,t)$ to solve the equation for $p$ (43).

3. Iterate the process, by replacing at each step either $p$ or $q$ by the newly obtained function. After a few iterations ($\sim 3$), it is usually helpful to replace the functions by a linear combination of the last two, e.g.,

$$q(x,t) \longleftarrow \alpha q_{\text{new}}(x,t) + (1-\alpha)q_{\text{old}}(x,t), \tag{D.1}$$

to avoid oscillating between two solutions. For instance, we use $\alpha = 0.75$.

4. Repeat until the difference between the last two solutions is small enough, which can be measured for instance by

$$\int [q_{\text{new}}(x,t) - q_{\text{old}}(x,t)]^2 \mathrm{d}x\mathrm{d}t. \tag{D.2}$$

To use standard methods for partial differential equations, we regularise the Heaviside $\Theta$ function by

$$\Theta(x) \simeq \frac{1 + \tanh(ax)}{2}, \tag{D.3}$$

with $a \sim 150$. This regularization causes a small discrepancy between the numerical solution and the exact one near the discontinuity of the function $q(x,1)$.

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
