# Peer review of "Joint distribution of currents in the symmetric exclusion process"

_SciPost Physics, doi:SciPost Phys. 16, 016 (2024)_

## Round 1 · Author Response

Warnings issued while processing user-supplied markup:
- Inconsistency: Markdown and reStructuredText syntaxes are mixed. Markdown will be used.
Add "#coerce:reST" or "#coerce:plain" as the first line of your text to force reStructuredText or no markup.
You may also contact the helpdesk if the formatting is incorrect and you are unable to edit your text.
Dear Editor,
We were happy to learn that all three Referees support publication of our article in SciPost Physics. We have followed the Referee's suggestions, and modified the manuscript accordingly. You can find below the list of changes, and our detailed responses to the Referees. The modifications appear in blue in the attached version of the manuscript.
We thank you in advance for your consideration,
Yours sincerely,
Aur\'elien Grabsch and Olivier B\'enichou
Response to Referee 1
We thank the Referee for their positive assessment of our manuscript, and for their suggestions to improve the readability of the article. We have implemented these suggestions in the revised version.
Concerning the general suggestions of the Referee:
-
and 2. We have followed the Referee's suggestions and clarified these points. In particular we now state explicitly the nature of the approaches used in these works (microscopic and/or macroscopic).
-
We have followed the Referee's suggestion and added a more detailed physical discussion of the profiles $\Phi$ and $\bar\Phi$ at the beginning of Section 2, when these objects are introduced. We have also moved Fig. 1 there to support this discussion.
Concerning the minor issues raised by the Referee:
-
We have modified this sentence.
-
We have added missing the description of Sections 6 and 7 in the introduction.
-
The regularisation is indeed not required to define the generalised current. Originally, we decided to introduce this generalisation, as we thought it made the definition more intuitive. In the revised version, we followed the Referee's suggestion to use a definition which does not involve any regularisation.
-
We have modified this point in the revised version
-
We have added the definitions of the error function and the Owen T function.
-
We have clarified which parts hold for $\rho_+ \neq \rho_-$ and which ones are specific to $\rho_+ = \rho_- = \rho$.
Response to Referee 2
We thank the Referee for their positive assessment of our manuscript, and for supporting publication in Scipost Physics. We have modified the article according to the suggestions of the Referee.
Concerning the main points raised by the Referee:
-
We have added a comparison with numerical simulations of the SEP in Section 2. Since the correlations between the density and the currents $Q_t$ or $J_t$ have already been studied independently in [6,7], we have added a comparison with our result on the joint correlation function $\langle \eta_r \: Q_t \: J_t \rangle_c$, now shown in Fig. 2. This figure shows an excellent agreement between our analytical result and the numerical simulations.
-
We have added Appendices C and D to describe the numerical methods used to solve (i) the integral equations and (ii) the MFT equations. Concerning the existence of solutions to the integral equations, we are not able to fully characterise the conditions for uniqueness of the solution. When $\alpha$ and $\beta$ are small, it is possible to construct a perturbative solution of these equations, which is unique. This is done in Section 7 to obtain the first cumulants and correlations. We have added a comment on this point in Section 5.
-
The $\delta$ functions appear not only because of the choice of the step initial condition, but also due to the observables under consideration. The study of $Q_t$ and $J_t$ introduce Heaviside step functions as final conditions for $p$, see Eq. (44) left. Then, the mapping to the AKNS equations yields these $\delta$ functions since it involves the derivative of $p$, even for a flat initial density. In principle, this could also be regularised by replacing the Heaviside step function by a smooth function. However, these discontinuities are not only present in the initial or final boundary conditions, but also in the solutions of the MFT equations themselves: the solution $q(x,1)$ becomes discontinuous at $x=0$ and $x=\xi$. To follow the Referee's suggestion to introduce a smoothing and then take a limit, one would need to know how the solution of the MFT equations behaves. But these MFT equations are the ones we aim to solve using the scattering technique. Therefore implementing this method does not seem easy, and we rely on introducing unknown constants multiplying the $\delta$ functions which are later determined by other means. Concerning the change of normalisation of $u$ and $v$ using the invariance of the AKNS equations, this is not a problem in our case, and does not affect our conclusions. Indeed, due to the $\delta$ singularities, we are only able to relate $u$ and $v$ at initial and final times to $\partial_x q(x,1)$ and $\partial_x q(x,0)$ up to some unknown proportionality constants, see Eqs.~(74,77). Therefore, changing the normalisation only redefines these constants. Since we only determine them at a later stage, these redefinitions are irrelevant.
Concerning the small comments raised by the Referee:
-
Indeed, we have added a comment below equation (5).
-
In this section we only present results for the case $\rho_+ = \rho_- = \rho$. We have added a sentence at the beginning of the section to specify this point.
-
This is indeed surprising. We do not have an understanding as to why this is the case. We have added a small comment in the revised version.
-
Indeed, we have added the expressions of $D$ and $\sigma$ for the SEP after the equation.
-
Thank you. We have now fixed this typo.
-
We have replaced this equation by a reference to Eq. (32) in the revised version.
-
Indeed, we have clarified the discussion below Eqs. (64-65) by moving to a footnote the discussion of the case in which time reversal symmetry is preserved (corresponding to the current $Q_t$ with an initial step density). This whole section indeed applies to arbitrary initial condition $\rho_0(x)$.
-
See the answer to point 3 above.
Response to Referee 3
We thank the Referee for their positive assessment of our manuscript, and for supporting publication in Scipost Physics. We have modified the article according to the suggestions of the Referee:
-
Thank you, we have corrected this typo.
-
We have added two appendices, C and D, which describe the numerical methods we use to solve the integral equations (8) and (9), as well as the method to solve numerically the MFT equations (42,43).
Concerning the perturbative solution, it is expected to agree well for small values of $\lambda$ and $\nu$ only, since we have computed explicitly only the first few orders. We have added a figure in Section 6 to compare it to the numerical solution of our integral equations. Remarkably, the perturbative solution is in good agreement even for values of $\lambda$ and $\nu$ which are not small ($\lambda=\nu=1)$, but ultimately differs for larger values.
-
This has been implemented in the revised version to clarify Eqs. (74,77).
-
The notation in (108) was indeed confusing. We have followed the Referee’s suggestion in the revised version.
-
We agree with the Referee that these relations can be obtained by different means. The condition (103) can indeed be obtained from the conservation law of the MFT. We have added a comment in the revised version.

---

## Round 1 · List of Changes

Two appendices have been added to describe the numerical methods that are used to solve (i) the integral equations and (ii) the coupled MFT equations.
We have added numerical simulations of the SEP (Fig. 2), and compared them to our analytical results.
We have clarified which parts of the manuscript hold for an arbitrary initial density of particles, and which ones are specific to the case of a flat initial density.
A new figure (Fig. 3) shows the comparison between the numerical solution of the integral equations and the perturbative solution computed in Section 6.
We have clarified several points raised by the Referees (see detailed responses to the Referees).
Finally we have corrected several typos.

---

## Editorial Decision

published